# MetaOptimize: A Framework for Optimizing Step Sizes and Other Meta-parameters

## Abstract

We address the challenge of optimizing meta-parameters (i.e., hyperparameters) in machine learning algorithms, a critical factor influencing training efficiency and model performance. Moving away from the computationally expensive traditional meta-parameter search methods, we introduce MetaOptimize framework that dynamically adjusts meta-parameters, particularly step sizes (also known as learning rates), during training. More specifically, MetaOptimize can wrap around any first-order optimization algorithm, tuning step sizes on the fly to minimize a specific form of regret that accounts for long-term effect of step sizes on training, through a discounted sum of future losses. We also introduce low complexity variants of MetaOptimize that, in conjunction with its adaptability to multiple optimization algorithms, demonstrate performance competitive to those of best hand-crafted learning rate schedules across various machine learning applications.

## 1 Introduction

Optimization algorithms used in machine learning involve meta-parameters (i.e., hyperparameters) that substantially influence their performance. These meta-parameters are typically identified through a search process, such as grid search or other trial-and-error methods, prior to training. However, the computational cost of this meta-parameter search is significantly larger than that of training with optimal meta-parameters (Dahl et al., 2023; Jin, 2022). Meta-parameter optimization seeks to streamline this process by concurrently adjusting meta-parameters during training, moving away from the computationally expensive and often sub-optimal trial and error search methods.

Meta-parameter optimization is particularly important in continual learning (De Lange et al., 2021), its primary domain, where dynamic environments or evolving loss functions necessitate meta-parameters, like step sizes, to adapt to optimal time-varying values rather than settling on a static value as in the stationary case. Nevertheless, this work concentrates on the stationary scenario, demonstrating the competitiveness of meta-parameter optimization even in this case.

In this work, we propose *MetaOptimize* as a framework for optimizing meta-parameters to minimize a form of regret, specifically accounting for the long-term influence of step sizes on future loss. The framework is applicable to a broad range of meta-parameters, however the primary focus of this paper is on step sizes as a critical meta-parameter that is universally present.

MetaOptimize brings additional benefits beyond simplifying the search process. Firstly, it enables a dynamic step-size adjustment during training, potentially accelerating the learning process. Traditional methods typically require manual customization of learning rate schedules for each problem, often following an optimal pattern of initial increase and subsequent decay (Amid et al., 2022). As our experiments show, step sizes obtained from MetaOptimize follow similar patterns automatically.

Secondly, varying step sizes across different blocks of a neural network, such as layers or neurons, has been shown to improve performance (Singh et al., 2015; Howard & Ruder, 2018). Manually tuning or using grid search for block-wise step-sizes is impractical in networks with numerous blocks. MetaOptimize framework can automatically manage blockwise step-sizes.

The concept of meta step-size optimization can be traced back to (Kesten, 1958), Delta-bar-Delta (Sutton, 1981; Jacobs, 1988), and its incremental variant, IDBD (Sutton, 1992). Over the years, numerous methods have been developed to address this challenge, detailed further in Section 8. This research distinguishes itself from prior work through the following key aspects:

- We introduce a formalization of step-size optimization as minimizing a specific form of regret, essentially a discounted sum of future losses. We demonstrate how to handle this minimization in a causal manner, by introducing the MetaOptimize framework.

- MetaOptimize framework is general in the sense that it can wrap around any first-order optimization algorithm, also called *base update*, (such as SGD, RMSProp (Hinton, 2012), Adam (Kingma & Ba, 2014), or Lion (Chen et al., 2023))), for which it optimizes step sizes via an algorithm of desire (such as SGD, Adam, RMSProp, or Lion), called the *meta update*.

- We develop approximation methods (Section 6), that when integrated into MetaOptimize, lead to computationally efficient algorithms that outperform state-of-the-art automatic hyperparameter optimization methods on CIFAR10, ImageNet, and language modeling applications (refer to experiments in Section 7).

- We show that some existing methods (like IDBD, its extension (Xu et al., 2018), and hypergradient descent (Baydin et al., 2017)) are specific instances or approximations within the MetaOptimize framework (see Section 5).

## 2 PROBLEM SETTING

We introduce a general continual optimization setting that, for a given sequence of loss functions $f_t(\cdot) : \mathbb{R}^n \to \mathbb{R}$, $t = 0, 1, 2, \ldots$, aims to find a sequence of weight vectors $\boldsymbol{w}_1, \boldsymbol{w}_2, \boldsymbol{w}_3, \ldots$ to minimize a discounted sum of future losses:

$$F_t^\gamma \overset{\text{def}}{=} (1 - \gamma) \sum_{\tau > t} \gamma^{\tau - t - 1} f_\tau(\boldsymbol{w}_\tau), \tag{1}$$

where $\gamma \in [0, 1)$ is a fixed constant, often very close to 1, called the *discount factor*. As an important special case, the above setting includes stationary supervised learning if $f_t$ are sampled from a static distribution, for all $t$. In this case, minimizing $F_t^\gamma$ results in rapid minimization of expected loss.

Consider an arbitrary first order optimization algorithm (including but not limited to SGD, RMSProp, Adam, or Lion) for updating $\boldsymbol{w}_t$. At each time $t$, this algorithm takes the gradient $\nabla f_t(\boldsymbol{w}_t)$ of the immediate loss function, along with an $m$-dimensional vector $\boldsymbol{\beta}_t$ of meta-parameters, and updates $\boldsymbol{w}_t$ and possibly some internal variables (e.g., momentum in Adam or trace of gradient squares in RMSProp), based on a fixed update rule $\text{Alg}_{\text{base}}$, referred to as the *base-update*,

$$\boldsymbol{x}_{t+1} = \text{Alg}_{\text{base}}(\boldsymbol{x}_t, \nabla f_t(\boldsymbol{w}_t), \boldsymbol{\beta}_t), \tag{2}$$

where $\boldsymbol{x}_t \overset{\text{def}}{=} \text{Stack}(\boldsymbol{w}_t, \tilde{\boldsymbol{x}}_t)$ is an $\tilde{n}$-dimensional vector obtained by stacking $\boldsymbol{w}_t$ and all internal variables of the algorithm that are being updated (e.g., momentum), denoted by $\tilde{\boldsymbol{x}}_t$. The goal of the MetaOptimize framework is to find a sequence of meta-parameters $\boldsymbol{\beta}_t$, for $t = 1, 2, \ldots$, such that when plugged into the base update, (2), results in relative minimization of $F_t^\gamma$ defined in (1).

Step-size optimization is a special case of the above framework where at each time $t$, the $m$ dimensional vector $\boldsymbol{\beta}_t$ is used to determine the $n$-dimensional (weight-wise) vector $\boldsymbol{\alpha}_t$ of step sizes (typically $m \ll n$), through a fixed function $\sigma : \mathbb{R}^m \to \mathbb{R}^n$,

$$\boldsymbol{\alpha}_t = \sigma(\boldsymbol{\beta}_t). \tag{3}$$

A typical choice is to partition weights of the neural network into $m$ blocks and use step-size $\exp(\beta)$ within each block for some entry $\beta$ of $\boldsymbol{\beta}$. Depending on $m$, this can result in a single shared scalar step-size, or layer-wise, node-wise, or weight-wise step sizes. It is particularly beneficial to consider a function $\sigma$ of the exponential form, mentioned above, because of two reasons (Sutton, 1992). First, it ensures that $\boldsymbol{\alpha}_t$ will always be positive. Second, a constant change in $\boldsymbol{\beta}_t$ would lead to a multiplicative change in $\boldsymbol{\alpha}_t$, making it suitable for adapting step sizes with different orders of magnitude.

## 3 FORWARD AND BACKWARD VIEWS

Since the definition of $F_t^\gamma$ in (1) relies on information forward into the future, minimizing it in a causal way necessitates alternative views; discussed in this section. In order to motivate our approach, we start by considering a hypothetical meta-parameter optimization algorithm that has oracle access

to future information (e.g., future loss), and updates $\boldsymbol{\beta}_t$ along the gradient of $F_t^{\gamma}$ with respect to $\boldsymbol{\beta}_t$; that is for $t = 0, 1, 2, \ldots$,

$$\boldsymbol{\beta}_{t+1} = \boldsymbol{\beta}_t - \eta \frac{\mathrm{d}}{\mathrm{d}\,\boldsymbol{\beta}_t} F_t^{\gamma} = \boldsymbol{\beta}_t - \eta \,(1 - \gamma) \sum_{\tau > t} \gamma^{\tau - t - 1} \frac{\mathrm{d}}{\mathrm{d}\,\boldsymbol{\beta}_t} f_\tau(\boldsymbol{w}_\tau), \tag{4}$$

for some fixed *meta step-size*, $\eta > 0$. This *forward-view* update however requires that at time $t$, we have access to $f_\tau(\cdot)$ and $\boldsymbol{w}_\tau$ for all $\tau > t$, which are typically unavailable. To circumvent this problem, we adopt an idea similar to *eligibility traces* in reinforcement learning (Sutton, 1988; Sutton & Barto, 2018). More specifically, instead of the forward-view update, we introduce an update of the following type, which we call the *backward-view* update. At time $\tau = 0, 1, 2, \ldots$, we let

$$\boldsymbol{\beta}_{\tau+1} \leftarrow \boldsymbol{\beta}_\tau - \eta \,(1 - \gamma) \sum_{t < \tau} \gamma^{\tau - t - 1} \frac{\mathrm{d}}{\mathrm{d}\,\boldsymbol{\beta}_t} f_\tau(\boldsymbol{w}_\tau). \tag{5}$$

Note that every term $\gamma^{\tau - t - 1} \frac{\mathrm{d}\, f_\tau(\boldsymbol{w}_\tau)}{\mathrm{d}\,\boldsymbol{\beta}_t}$ in the right hand side of (4) also appears in (5), but is applied at time $\tau$ instead of time $t$, which is the earliest time that all required information for computing this term is available. Similar to the eligibility traces in RL, backward view updates are accurate approximation of the forward view updates for sufficiently small meta-step sizes (i.e., when $\eta \to 0$), in the following sense: consider some $T \geq 1$ and suppose that $f_t(\cdot) = 0$ for all $t < 0$ and all $t > T$. Then, as $\eta \to 0$, it can be shown that $(\beta_T^{(5)} - \beta_0)/\eta \to (\beta_T^{(4)} - \beta_0)/\eta$, where $\beta_T^{(5)}$ and $\beta_T^{(4)}$ are the values of $\beta$ at time $T$ obtained from updates (5) and (4), respectively, starting from the same initial value $\beta_0$ at time 0. This is because as $\eta \to 0$, $\beta$ remains almost constant over the interval $[0, T]$, and the right hand side of (5) would be equal to the right hand side of (4) when summed over $[0, T]$, with accuracy $O(\eta^2)$. For larger values of $\eta$, the approximation may not be as accurate. Refer to Section 9 for a discussion on more accurate approximations.

In light of (5), the $\widehat{\nabla_{\boldsymbol{\beta}} F}_\tau$ defined below serves as a causal proxy for $\mathrm{d}\, F_\tau^{\gamma}/\mathrm{d}\,\boldsymbol{\beta}_\tau$;

$$\widehat{\nabla_{\boldsymbol{\beta}} F}_\tau \stackrel{\text{def}}{=} (1 - \gamma) \sum_{t=0}^{\tau-1} \gamma^{\tau - t - 1} \frac{\mathrm{d}}{\mathrm{d}\,\boldsymbol{\beta}_t} f_\tau(\boldsymbol{w}_\tau). \tag{6}$$

It follows from chain rule that

$$\widehat{\nabla_{\boldsymbol{\beta}} F}_\tau = \mathcal{H}_\tau^T \nabla f_\tau(\boldsymbol{w}_\tau), \tag{7}$$

where

$$\mathcal{H}_\tau \stackrel{\text{def}}{=} (1 - \gamma) \sum_{t=0}^{\tau-1} \gamma^{\tau - t - 1} \frac{d\boldsymbol{w}_\tau}{\mathrm{d}\,\boldsymbol{\beta}_t}. \tag{8}$$

The $d\boldsymbol{w}_\tau/\mathrm{d}\,\boldsymbol{\beta}_t$ in (8) denotes the Jacobian matrix of $\boldsymbol{w}_\tau$ with respect to $\boldsymbol{\beta}_t$. Therefore, $\mathcal{H}_\tau$ is an $n \times m$ matrix such that $\mathcal{H}_\tau \, \boldsymbol{v}$, for any $m \times 1$ vector $\boldsymbol{v}$, equals the change in $\boldsymbol{w}_\tau$ if we increment all past $\boldsymbol{\beta}_t$ along $\gamma^{\tau - t} \boldsymbol{v}$, while taking into account the non-linear dynamics of $\boldsymbol{\beta}$ (i.e., the impact of each $\boldsymbol{\beta}_t$ increment on $\boldsymbol{\beta}_\tau$ of future times $\tau > t$).

## 4 METAOPTIMIZE

The general formulation of MetaOptimize framework is given in Algorithm 1. The idea is to update $\boldsymbol{\beta}_t$ via any first order optimization algorithm to minimize $F_t^{\gamma}$, while using the surrogate gradient $\widehat{\nabla_{\boldsymbol{\beta}} F}_t$ in place of $\nabla_{\boldsymbol{\beta}} F_t^{\gamma}$, to preserve causality of the updates. More specifically, for $t = 1, 2, \ldots$, let

$$\boldsymbol{y}_{t+1} = \mathrm{Alg}_{\mathrm{meta}} \left( \boldsymbol{y}_t, \widehat{\nabla_{\boldsymbol{\beta}} F}_t \right) = \mathrm{Alg}_{\mathrm{meta}} \left( \boldsymbol{y}_t, \mathcal{H}_t^T \nabla f_t(\boldsymbol{w}_t) \right) \tag{9}$$

be the *meta update*, where $\boldsymbol{y}_t \stackrel{\text{def}}{=} \mathrm{Stack}(\boldsymbol{\beta}_t, \tilde{\boldsymbol{y}}_t)$ is an $\tilde{m}$-dimensional vector obtained from stacking $\boldsymbol{\beta}_t$ and all other internal variables $\tilde{\boldsymbol{y}}_t$ of the $\mathrm{Alg}_{\mathrm{meta}}$ algorithm (e.g., momentum), and the second equality follows from (7). Examples of $\mathrm{Alg}_{\mathrm{meta}}$ include SGD, RMSprop, Adam, and Lion algorithms. Note that in all cases, we pass $\widehat{\nabla_{\boldsymbol{\beta}} F}$ to the algorithm as the gradient.

In each iteration, after performing the base update (2), we compute $\mathcal{H}_t^T \nabla f_t(\boldsymbol{w}_t)$ and plug it into (9) to update $\boldsymbol{y}$ (and in particular $\boldsymbol{\beta}$). In the rest of this section, we present incremental updates for $\mathcal{H}_t$.

Let $\boldsymbol{h}_t$ be an $nm$-dimensional vector obtained by stacking the columns of the $n \times m$ matrix $\mathcal{H}_t$. It follows from the chain rule that for any times $t$ and $\tau$ with $t \geq \tau$,

$$\frac{\mathrm{d}\,\boldsymbol{y}_{t+1}}{\mathrm{d}\,\boldsymbol{\beta}_\tau} = \frac{\mathrm{d}\,\boldsymbol{y}_{t+1}}{\mathrm{d}\,\boldsymbol{y}_t}\frac{\mathrm{d}\,\boldsymbol{y}_t}{\mathrm{d}\,\boldsymbol{\beta}_\tau} + \frac{\mathrm{d}\,\boldsymbol{y}_{t+1}}{\mathrm{d}\,\boldsymbol{x}_t}\frac{\mathrm{d}\,\boldsymbol{x}_t}{\mathrm{d}\,\boldsymbol{\beta}_\tau} + \frac{\mathrm{d}\,\boldsymbol{y}_{t+1}}{\mathrm{d}\,\boldsymbol{h}_t}\frac{\mathrm{d}\,\boldsymbol{h}_t}{\mathrm{d}\,\boldsymbol{\beta}_\tau},$$

$$\frac{\mathrm{d}\,\boldsymbol{x}_{t+1}}{\mathrm{d}\,\boldsymbol{\beta}_\tau} = \frac{\mathrm{d}\,\boldsymbol{x}_{t+1}}{\mathrm{d}\,\boldsymbol{y}_t}\frac{\mathrm{d}\,\boldsymbol{y}_t}{\mathrm{d}\,\boldsymbol{\beta}_\tau} + \frac{\mathrm{d}\,\boldsymbol{x}_{t+1}}{\mathrm{d}\,\boldsymbol{x}_t}\frac{\mathrm{d}\,\boldsymbol{x}_t}{\mathrm{d}\,\boldsymbol{\beta}_\tau} + \frac{\mathrm{d}\,\boldsymbol{x}_{t+1}}{\mathrm{d}\,\boldsymbol{h}_t}\frac{\mathrm{d}\,\boldsymbol{h}_t}{\mathrm{d}\,\boldsymbol{\beta}_\tau},$$

$$\frac{\mathrm{d}\,\boldsymbol{h}_{t+1}}{\mathrm{d}\,\boldsymbol{\beta}_\tau} = \frac{\mathrm{d}\,\boldsymbol{h}_{t+1}}{\mathrm{d}\,\boldsymbol{y}_t}\frac{\mathrm{d}\,\boldsymbol{y}_t}{\mathrm{d}\,\boldsymbol{\beta}_\tau} + \frac{\mathrm{d}\,\boldsymbol{h}_{t+1}}{\mathrm{d}\,\boldsymbol{x}_t}\frac{\mathrm{d}\,\boldsymbol{x}_t}{\mathrm{d}\,\boldsymbol{\beta}_\tau} + \frac{\mathrm{d}\,\boldsymbol{h}_{t+1}}{\mathrm{d}\,\boldsymbol{h}_t}\frac{\mathrm{d}\,\boldsymbol{h}_t}{\mathrm{d}\,\boldsymbol{\beta}_\tau}.$$

Letting

$$G_t \stackrel{\text{def}}{=} \begin{bmatrix} \frac{\mathrm{d}\,\boldsymbol{y}_{t+1}}{\mathrm{d}\,\boldsymbol{y}_t} & \frac{\mathrm{d}\,\boldsymbol{y}_{t+1}}{\mathrm{d}\,\boldsymbol{x}_t} & \frac{\mathrm{d}\,\boldsymbol{y}_{t+1}}{\mathrm{d}\,\boldsymbol{h}_t} \\ \frac{\mathrm{d}\,\boldsymbol{x}_{t+1}}{\mathrm{d}\,\boldsymbol{y}_t} & \frac{\mathrm{d}\,\boldsymbol{x}_{t+1}}{\mathrm{d}\,\boldsymbol{x}_t} & \frac{\mathrm{d}\,\boldsymbol{x}_{t+1}}{\mathrm{d}\,\boldsymbol{h}_t} \\ \frac{\mathrm{d}\,\boldsymbol{h}_{t+1}}{\mathrm{d}\,\boldsymbol{y}_t} & \frac{\mathrm{d}\,\boldsymbol{h}_{t+1}}{\mathrm{d}\,\boldsymbol{x}_t} & \frac{\mathrm{d}\,\boldsymbol{h}_{t+1}}{\mathrm{d}\,\boldsymbol{h}_t} \end{bmatrix}, \tag{10}$$

the above set of equations can be equivalently written as

$$\left[\frac{\mathrm{d}\,\boldsymbol{y}_{t+1}}{\mathrm{d}\,\boldsymbol{\beta}_\tau}\ \frac{\mathrm{d}\,\boldsymbol{x}_{t+1}}{\mathrm{d}\,\boldsymbol{\beta}_\tau}\ \frac{\mathrm{d}\,\boldsymbol{h}_{t+1}}{\mathrm{d}\,\boldsymbol{\beta}_\tau}\right]^T = G_t \left[\frac{\mathrm{d}\,\boldsymbol{y}_t}{\mathrm{d}\,\boldsymbol{\beta}_\tau}\ \frac{\mathrm{d}\,\boldsymbol{x}_t}{\mathrm{d}\,\boldsymbol{\beta}_\tau}\ \frac{\mathrm{d}\,\boldsymbol{h}_t}{\mathrm{d}\,\boldsymbol{\beta}_\tau}\right]^T.$$

It follows that

$$\sum_{\tau=0}^{t} \gamma^{t-\tau} \left[\frac{\mathrm{d}\,\boldsymbol{y}_{t+1}}{\mathrm{d}\,\boldsymbol{\beta}_\tau}\ \frac{\mathrm{d}\,\boldsymbol{x}_{t+1}}{\mathrm{d}\,\boldsymbol{\beta}_\tau}\ \frac{\mathrm{d}\,\boldsymbol{h}_{t+1}}{\mathrm{d}\,\boldsymbol{\beta}_\tau}\right]^T = G_t \left[\frac{\mathrm{d}\,\boldsymbol{y}_t}{\mathrm{d}\,\boldsymbol{\beta}_t}\ \frac{\mathrm{d}\,\boldsymbol{x}_t}{\mathrm{d}\,\boldsymbol{\beta}_t}\ \frac{\mathrm{d}\,\boldsymbol{h}_t}{\mathrm{d}\,\boldsymbol{\beta}_t}\right]^T + G_t \sum_{\tau=0}^{t-1} \gamma^{t-\tau} \left[\frac{\mathrm{d}\,\boldsymbol{y}_t}{\mathrm{d}\,\boldsymbol{\beta}_\tau}\ \frac{\mathrm{d}\,\boldsymbol{x}_t}{\mathrm{d}\,\boldsymbol{\beta}_\tau}\ \frac{\mathrm{d}\,\boldsymbol{h}_t}{\mathrm{d}\,\boldsymbol{\beta}_\tau}\right]^T. \tag{11}$$

Let

$$Y_t \stackrel{\text{def}}{=} (1-\gamma) \sum_{\tau=0}^{t-1} \gamma^{t-\tau-1} \frac{\mathrm{d}\,\boldsymbol{y}_t}{\mathrm{d}\,\boldsymbol{\beta}_\tau} \tag{12}$$

$$X_t \stackrel{\text{def}}{=} (1-\gamma) \sum_{\tau=0}^{t-1} \gamma^{t-\tau-1} \frac{\mathrm{d}\,\boldsymbol{x}_t}{\mathrm{d}\,\boldsymbol{\beta}_\tau}, \tag{13}$$

$$Q_t \stackrel{\text{def}}{=} (1-\gamma) \sum_{\tau=0}^{t-1} \gamma^{t-\tau-1} \frac{\mathrm{d}\,\boldsymbol{h}_t}{\mathrm{d}\,\boldsymbol{\beta}_\tau}. \tag{14}$$

Note also that $\mathrm{d}\,\boldsymbol{x}_t/\mathrm{d}\,\boldsymbol{\beta}_t = 0$, $\mathrm{d}\,\boldsymbol{h}_t/\mathrm{d}\,\boldsymbol{\beta}_t = 0$, and $\mathrm{d}\,\boldsymbol{y}_t/\mathrm{d}\,\boldsymbol{\beta}_t = \mathrm{d}\,\mathrm{Stack}(\boldsymbol{\beta}_t, \tilde{\boldsymbol{y}}_t)/\mathrm{d}\,\boldsymbol{\beta}_t = \mathrm{Stack}(I, 0)$. Plugging these into (11), we obtain

$$\begin{bmatrix} Y_{t+1} \\ X_{t+1} \\ Q_{t+1} \end{bmatrix} = G_t \left( \gamma \begin{bmatrix} Y_t \\ X_t \\ Q_t \end{bmatrix} + (1-\gamma) \begin{bmatrix} \begin{bmatrix} I \\ 0 \end{bmatrix} \\ 0 \\ 0 \end{bmatrix} \right). \tag{15}$$

Matrices $X_t, Y_t, Q_t$ can be computed iteratively using (15). The matrix $\mathcal{H}_t$ in (8) is then obtained from the sub-matrix constituting the first $n$ rows of $X_t$, because $\boldsymbol{x}_t = \mathrm{Stack}(\boldsymbol{w}_t, \tilde{\boldsymbol{x}}_t)$.

To complete Algorithm 1, it only remains to compute the matrix $G_t$ in (10). In Appendix A, we calculate $G_t$ for common choices of base and meta updates: SGD, AdamW, and Lion. Notably, the first row of $G_t$ blocks depends only on $\mathrm{Alg}_{\text{meta}}$, and the rest of $G_t$ blocks depend only on $\mathrm{Alg}_{\text{base}}$. This simplifies the derivation and implementation for various base and meta algorithm combinations.

*Remark* 4.1. A key distinction of MetaOptimize from existing meta-parameter optimization methods is that it accounts for the dynamics of the meta-parameters $\boldsymbol{\beta}$—specifically, how changes in the current $\boldsymbol{\beta}$ affect future values of $\boldsymbol{\beta}$. This is captured by the $Y_t$ matrix defined in (12), whose influence then propagates into $\mathcal{H}_t$ and the meta-update (see (15)). To provide more intuition, lets focus on a simple case with one-dimensional $\beta$ and SGD meta-updates, and consider two cases: a) If $\beta_t$ has consistently increased over the recent past trying to track the optimal $\beta$, then $Y_t$ will grow large, resulting in significant increments of $H_t$. This increases the norm of $H_t$, and improves the tracking of optimal $\beta$. b) If $\beta_t$ has remained nearly constant, suggesting convergence to the optimal value, $Y_t$ will shrink, leading to smaller $H_t$ increments and smaller updates to $\beta_t$. This helps stabilize $\beta$ around its optimal value.

---

**Algorithm 1**    MetaOptimize Framework   (for general meta-parameters)

---

**Given:** Base-update $\text{Alg}_{\text{base}}$, meta-update $\text{Alg}_{\text{meta}}$,
**Parameters:** Discount-factor $\gamma \leq 1$.
**Initialize:** $X_0 = 0_{(n+\tilde{n}) \times m}$, $Y_0 = \left[ I_{m \times m} \mid 0_{m \times \tilde{m}} \right]^T$, and $Q_0 = 0_{nm \times m}$.
**for** $t = 0, 1, 2, \ldots$ **do**
$\quad \boldsymbol{x}_{t+1} \leftarrow \text{Alg}_{\text{base}}(\boldsymbol{x}_t, \nabla f_t(\boldsymbol{w}_t), \boldsymbol{\beta}_t)$.
$\quad \mathcal{H}_t = $ sub-matrix of $X_t$, constituting its first $n$ rows.
$\quad \boldsymbol{y}_{t+1} \leftarrow \text{Alg}_{\text{meta}} \left( \boldsymbol{y}_t, \mathcal{H}_t^T \nabla f_t(\boldsymbol{w}_t) \right)$.
$\quad$ Update $X_t$, $Y_t$, and $Q_t$ from (15), using $G_t$ in (10).
**end for**

---

## 5   REDUCING COMPLEXITY

The matrix $G_t$ in (10) is typically large, reducing the algorithm's practicality. We discuss two approximations of $G_t$ for more efficient algorithms.

**2×2 approximation:** The vector $\boldsymbol{h}_t$, formed by stacking $\mathcal{H}_t$'s columns, has length $mn$, making $G_t$'s last row and column of blocks very large. Moreover, as shown in Appendix A, the term $d\boldsymbol{h}_{t+1}/d\boldsymbol{x}_t$ typically involves third order derivatives of $f_t$ with respect to $\boldsymbol{w}_t$, which is not practically computable.

In the 2×2 approximation, we resolve the above problems by completely zeroing out all blocks in the last row and also in the last column of blocks of $G_t$ in (10). Consequently, we can also remove $Q_t$ from the algorithm. This appears to have minimal impact on the performance, as we empirically observed in simple settings. Intuitively, the block $\mathrm{d}\,\boldsymbol{x}_{t+1}/\mathrm{d}\,\boldsymbol{h}_t$ in $G_t$ is zero, as $\mathcal{H}_t$ doesn't affect the base update (2). Thus, $Q$ affects $X$, only indirectly, via $Y$.

**L-approximation:**  Herein, we take a step further, and in addition to the last row and the last column of blocks of $G_t$, we also zero out the block in the first row and the second column of $G_t$. In other words, we let

$$G_t^L \overset{\text{def}}{=} \begin{bmatrix} \frac{\mathrm{d}\,\boldsymbol{y}_{t+1}}{\mathrm{d}\,\boldsymbol{y}_t} & 0 \\ \frac{\mathrm{d}\,\boldsymbol{x}_{t+1}}{\mathrm{d}\,\boldsymbol{y}_t} & \frac{\mathrm{d}\,\boldsymbol{x}_{t+1}}{\mathrm{d}\,\boldsymbol{x}_t} \end{bmatrix}, \tag{16}$$

and simplify (15) as

$$\begin{bmatrix} Y_{t+1} \\ X_{t+1} \end{bmatrix} = G_t^L \left( \gamma \begin{bmatrix} Y_t \\ X_t \end{bmatrix} + (1 - \gamma) \begin{bmatrix} I \\ 0 \\ \hline 0 \end{bmatrix} \right). \tag{17}$$

We have empirically observed that the resulting algorithm typically performs as good as the 2×2 approximation, and even results in improved stability in some cases.

**Intuition of MetaOptimize updates:** Algorithm 2 provides a 2×2 approximation of MetaOptimize for the case where both base and meta updates use SGD, and under scalar step-size (detailed derivation in Appendix A). It shows that $\mathcal{H}_t$ traces past gradients, decaying at rate $\gamma(I - [\boldsymbol{\alpha}] \nabla^2 f_t)$. This decay ensures that if past gradients poorly approximate future ones due to large $\nabla^2 f_t$ or $\boldsymbol{\alpha}$, their influence fades more rapidly. If the current gradient aligns positively with past gradients (i.e., $-\mathcal{H}_t^T \nabla f_t > 0$), the algorithm increases the step-size $\boldsymbol{\alpha}$ for quicker adaptation; if negatively correlated, it reduces the step size to prevent issues like zigzagging. $Y_t$ in (12) reflects the impact of changes in past $\boldsymbol{\beta}$ on the current value of $\boldsymbol{\beta}$, amplifying the increment in the $\mathcal{H}_{t+1}$ update if $\boldsymbol{\beta}$ has been consistently rising or falling over the recent past. It is also worth noting that in Algorithm 2, under the L-approximation, $Y_t$ remains constant, equal to $I$. A similar phenomenon occurs also when Adam, RMSProp, or Lion algorithms are used instead of SGD.

**Containing some existing algorithms as special cases:**  Special cases of the above L-approximation method include IDBD algorithm (Sutton, 1982) and its extension (Xu et al., 2018), if we limit $\text{Alg}_{\text{base}}$ and $\text{Alg}_{\text{meta}}$ to SGD algorithm. Refer to Appendix B.1 for more details and proofs.

MetaOptimize also contains the hypergradient-descent algorithm (Baydin et al., 2017) as a special case, when using SGD for both base and meta updates of MetaOptimze with $\gamma = 0$. Hypergradient-descent updates step size towards minimizing the immediate loss $f_t$ rather than discounted sum of future losses, $F_t^\gamma$, ignoring long-term effects of step size on future loss. See Appendix B.2 for details.

---

**Algorithm 2** MetaOptimize with $2\times2$ approx., $(\text{Alg}_{\text{base}}, \text{Alg}_{\text{meta}}) = (\text{SGD}, \text{SGD})$, and scalar step-size

    **Initialize:** $\mathcal{H}_0 = \mathbf{0}_{n\times 1}, Y_0 = 1$.
    **for** $t = 1, 2, \ldots$ **do**
       $\alpha_t = e^{\beta_t}$
       **Base update:**
          $\boldsymbol{w}_{t+1} = \boldsymbol{w}_t - \alpha_t \nabla f_t(\boldsymbol{w}_t)$
          $\mathcal{H}_{t+1} = \gamma\big(I - \alpha_t \nabla^2 f_t(\boldsymbol{w}_t)\big)\mathcal{H}_t - Y_t \alpha_t \nabla f_t(\boldsymbol{w}_t)$
          $Y_{t+1} = \gamma Y_t + (1-\gamma) - \gamma\eta \mathcal{H}_t^T \nabla^2 f_t(\boldsymbol{w}_t)\mathcal{H}_t$     `# For L-approximation let` $Y_{t+1} = 1$
       **Meta update:**
          $\beta_{t+1} = \beta_t - \eta\, \mathcal{H}_t^T \nabla f_t(\boldsymbol{w}_t)$
    **end for**

---

## 6   HESSIAN-FREE METAOPTIMIZE

The step-size optimization algorithms discussed so far typically involve Hessian, $\nabla^2 f_t(\boldsymbol{w}_t)$, of the loss function. In particular, the Hessian matrix typically appears in the middle column of blocks in the $G_t$ matrix; e.g., in the $d\boldsymbol{w}_{t+1}/d\boldsymbol{w}_t$ block where $\boldsymbol{w}_{t+1} = \boldsymbol{w}_t - \alpha_t \nabla f_t(\boldsymbol{w}_t)$. Consequently, the update in (15) involves a Hessian-matrix-product of the form $\nabla^2 f_t(\boldsymbol{w}_t)\mathcal{H}_t$, which increases per-step computational complexity of the algorithm. The added computational overhead would be still manageable if $m$ is small. In particular for $m = 1$ (i.e., the case that a scalar step-size is used for update of all weights), $\mathcal{H}_t$ would be a vector; and one can leverage efficient Hessian-vector-product computation techniques that have the same complexity as gradient computation (Pearlmutter, 1994).

Interestingly, for certain base and meta algorithms, we can eliminate the Hessian without much compromising the performance. An example of such (base or meta) algorithms is the Lion algorithm (Chen et al., 2023). The Lion algorithm, when used as the base algorithm, updates $\boldsymbol{w}_t$ as

$$\boldsymbol{m}_{t+1} = \rho\,\boldsymbol{m}_t + (1-\rho)\,\nabla f_t(\boldsymbol{w}_t),$$
$$\boldsymbol{w}_{t+1} = \boldsymbol{w}_t - \boldsymbol{\alpha}_t \operatorname{Sign}\big(c\,\boldsymbol{m}_t + (1-c)\nabla f_t\big) - \kappa \boldsymbol{\alpha}_t \boldsymbol{w}_t,$$

where $\rho, c \in [0,1)$, $\kappa$ is a nonnegative weight-decay parameter, and $\operatorname{Sign}(\cdot)$ is the entry-wise sign function. In the special cases of $c = 0$ or $\rho = 0$, $\boldsymbol{m}_t$ can be eliminated and the above update simplifies to $\boldsymbol{w}_{t+1} = \boldsymbol{w}_t - \boldsymbol{\alpha}_t \operatorname{Sign}(\nabla f_t) - \kappa \boldsymbol{\alpha}_t \boldsymbol{w}_t$. In this case, it is easy to see that the derivatives of $\boldsymbol{x}_t$ in (10) are Hessian-free. The above argument can be extended to arbitrary values of $c$ and $\rho$. In Appendix A.1.3 (respectively Appendix A.3.2), we show that if $\text{Alg}_{\text{meta}}$ ($\text{Alg}_{\text{base}}$) is the Lion algorithm, then the first row (second and third rows) of blocks in $G$ would be Hessian-free. In summary, Algorithm 1 turns Hessian-free, if Lion is used in both base and meta updates. This elimination of Hessian results from flatness of the Sign function when ignoring the discontinuity at 0.

For other algorithms, we may consider their *Hessian-free approximation* by zeroing out any Hessian term in $G_t$. The Hessian-free approximation turns out to be a good approximation, especially for base and meta algorithms that involve gradient normalization, like RMSProp and Adam. Note that, the sign function used in the Lion algorithm is an extreme form of normalization that divides a vector by its absolute value. We could instead use softer forms of normalization, such as normalizing to square root of a trace of squared vector, $\boldsymbol{v}_t$, as in RMSProp. Such normalizations typically result in two opposing Hessian-based terms in $\mathcal{H}_t$'s update (stemming from $\frac{\mathrm{d}\,\boldsymbol{w}_{t+1}}{\mathrm{d}\,\boldsymbol{w}_t}$ and $\frac{\mathrm{d}\,\boldsymbol{w}_{t+1}}{\mathrm{d}\,\boldsymbol{v}_t}$ blocks of matrix $G_t$), aiming to cancel out, particularly when consecutive gradients are positively correlated.

The main advantage of Hessian-free methods lies in their computational congeniality. For base and meta updates including SGD, RMSProp, AdamW, and Lion, the Hessian-free $2\times2$ approximation has low computational complexity, requiring only a few vector-products per iteration beyond the computations required for the base and meta updates. When Hessian terms in $2 \times 2$ approximation of $G_t$ are zeroed out, the blocks in $G_t$, and therefore the blocks in $X_t$ and $Y_t$, become diagonal. Thus, $X_t$ and $Y_t$ matrices can be simplified to vector forms, eliminating costly matrix multiplications. The same holds for general blockwise step-sizes (e.g., layer-wise and weight-wise step-sizes), leading to computational overheads on par with the scalar case. We note also that for the meta updates mentioned above if we use no weight-decay in the meta update, Hessian-free $2\times2$ approximation becomes equivalent to Hessian-free L-approximation. Algorithm 3 presents Hessian-free approximations for some selected base and meta updates: *SGD with momentum (SGDm)*, AdamW, and Lion.

---

**Algorithm 3** Hessian-free MetaOptimize algorithms with $2\times2$ approximation used in experiments

---

**Parameters:** $\eta > 0$ (default $10^{-3}$), $\gamma \in [0, 1]$ (default 1)
**Initialize:** $\boldsymbol{h}_0 = \boldsymbol{0}_{n\times1}$.
**for** $t = 1, 2, \ldots$ **do**

> **Base update**
>
> $\quad\boldsymbol{\alpha}_t = \sigma(\boldsymbol{\beta}_t)$    # exponential scalar/blockwise
> $\quad\boldsymbol{m}_{t+1} = \rho\boldsymbol{m}_t + (1-\rho)\nabla f_t(\boldsymbol{w}_t)$
> $\quad$**if** $\text{Alg}_{\text{base}}$ is SGDm **then** $\quad\Delta\boldsymbol{w} = -\boldsymbol{\alpha}_t\boldsymbol{m}_t - \kappa\boldsymbol{\alpha}_t\boldsymbol{w}_t$
> $\quad$**if** $\text{Alg}_{\text{base}}$ is Lion **then** $\quad\Delta\boldsymbol{w} = -\boldsymbol{\alpha}_t\,\text{Sign}\left(c\,\boldsymbol{m}_t + (1-c)\nabla f_t\right) - \kappa\boldsymbol{\alpha}_t\boldsymbol{w}_t$
> $\quad$**if** $\text{Alg}_{\text{base}}$ is AdamW **then** $\quad\boldsymbol{v}_{t+1} = \lambda\,\boldsymbol{v}_t + (1-\lambda)\nabla f_t(\boldsymbol{w}_t)^2$
> $\qquad\qquad\qquad\qquad\qquad\quad\;\mu_t = \sqrt{1-\lambda^t}/(1-\rho^t),$
> $\qquad\qquad\qquad\qquad\qquad\quad\;\Delta\boldsymbol{w} = -\boldsymbol{\alpha}_t\mu_t\boldsymbol{m}_t/\sqrt{\boldsymbol{v}_t} - \kappa\boldsymbol{\alpha}_t\boldsymbol{w}_t$
> $\quad\boldsymbol{w}_{t+1} = \boldsymbol{w}_t + \Delta\boldsymbol{w}$
> $\quad\boldsymbol{h}_{t+1} = \gamma(1-\kappa\boldsymbol{\alpha}_t)\boldsymbol{h}_t + \Delta\boldsymbol{w}$

> **Meta update**
>
> $\quad\boldsymbol{z} = \boldsymbol{h}_t\,\nabla f_t(\boldsymbol{w}_t)$
> $\quad\bar{\boldsymbol{m}}_{t+1} = \bar{\rho}\,\bar{\boldsymbol{m}}_t + (1-\bar{\rho})\,\boldsymbol{z}$
> $\quad$**if** $\text{Alg}_{\text{meta}}$ is Lion **then** $\quad\boldsymbol{\beta}_{t+1} = \boldsymbol{\beta}_t - \eta\,\text{Sign}\left(\bar{c}\,\bar{\boldsymbol{m}}_t + (1-\bar{c})\boldsymbol{z}\right)$
> $\quad$**if** $\text{Alg}_{\text{meta}}$ is Adam **then** $\quad\bar{\boldsymbol{v}}_{t+1} = \bar{\lambda}\,\bar{\boldsymbol{v}}_t + (1-\bar{\lambda})\,\boldsymbol{z}^2$
> $\qquad\qquad\qquad\qquad\qquad\quad\;\bar{\mu}_t = \sqrt{1-\bar{\lambda}^t}/(1-\bar{\rho}^t)$
> $\qquad\qquad\qquad\qquad\qquad\quad\;\boldsymbol{\beta}_{t+1} = \boldsymbol{\beta}_t - \eta\,\bar{\mu}_t\bar{\boldsymbol{m}}_t/\sqrt{\bar{\boldsymbol{v}}_t}$

**end for**

---

# 7 EXPERIMENTS

In this section, we evaluate the MetaOptimize framework on image classification and language modeling benchmarks. Out of several possible combinations of base and meta algorithms and approximations, we report a few Hessian-free combinations from Algorithm 3 that showed better performance. In all experiments, we set the initial step-sizes of MetaOptimize to one or two orders of magnitudes smaller than the range of good fixed step-sizes, with no specific tuning. We compare MetaOptimize against some popular baselines whose meta-parameters are well-tuned for each task separately. Refer to Appendix C for further experiment details. Codes are available at (Anonymous, 2024).

## 7.1 CIFAR10 DATASET

The first set of experiments involve training ResNet-18 with batch size of 100 on the CIFAR10 (Krizhevsky et al., 2009) dataset. Fig. 1 depicts the learning curves of four combinations of (base, meta) algorithms for Hessian-free MetaOptimize, along with the corresponding baselines with well-tuned fixed step sizes. For MetaOptimize, in addition to scalar step-sizes, we also considered block-wise step-sizes by partitioning layers of the ResNet18 network into six blocks (first and last linear blocks and 4 ResNet blocks). Fig. 1 demonstrates that each tested base-meta combination of MetaOptimize, whether scalar or blockwise, surpasses the performance of the corresponding fixed step-size baseline.

Interestingly, as demonstrated in Fig. 2, the MetaOptimize algorithms show remarkable robustness to initial step-size choices, even for initial step sizes that are several orders of magnitude smaller than the optimal fixed step-size.

Fig. 3 depicts the blockwise step-sizes for (SGDm, Adam) across different blocks, showing an increasing trend from the first to the last block (output layer), which is generally a desirable pattern. In contrast, in the blockwise versions of (AdamW, Adam), (Lion, Lion), and (RMSProp, Adam) updates, we empirically observed that the first five blocks exhibit similar trends and values, while the last block follows a distinct trend, growing larger and rising at a later time.

## 7.2 IMAGENET DATASET

We trained ResNet-18 with batch-size 256 on ImageNet (Deng et al., 2009). We compared MetaOptimize with scalar step-size against four state-of-the-art hyperparamter optimization algorithms,

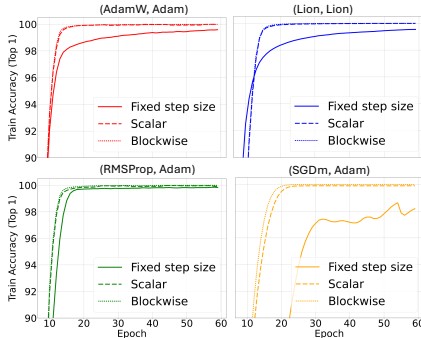

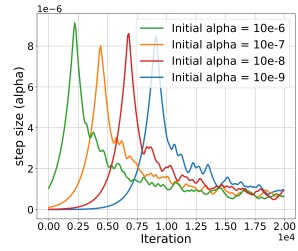

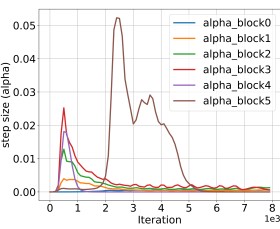

Figure 1: Learning curves for selected (base, meta) combinations in CIFAR10.

Figure 2: Robustness to initial step-sizes, for (Lion, Lion) as (base, meta) update in CIFAR10.

Figure 3: Evolution of blockwise step-sizes during training, for (SGDm, Adam) as (base, meta) update in CIFAR10.

namely DoG (Ivgi et al., 2023), gdtuo (Chandra et al., 2022), Prodigy (Mishchenko & Defazio, 2023), and mechanic (Cutkosky et al., 2024), as well as AdamW and Lion baselines with fixed step-sizes, and AdamW with a well-tuned cosine decay learning rate scheduler with a 10k iterations warmup. Learning curves and complexity overheads are shown respectively in Fig. 4 and Table 1, showcasing the advantage of MetaOptimize algorithms (learning curve of DoG is not depicted due to its relatively poor performance). Unlike CIFAR10, here the blockwise versions of MetaOptimize showed no improvement over the scalar versions. Refer to Appendix D for further details.

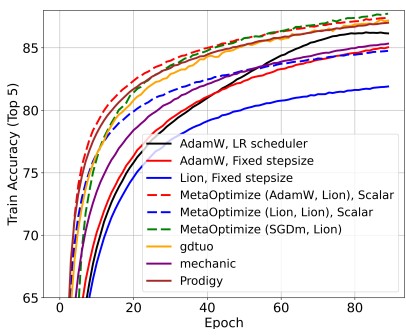

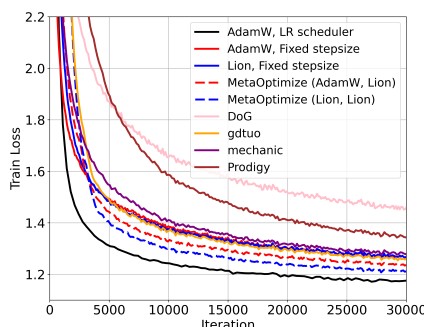

Figure 4: ImageNet learning curves.

Figure 5: TinyStories learning curves.

Table 1: Per-iteration wall-clock-time and GPU-space overhead (compared to AdamW).

|  | **ImageNet** | | **TinyStories** | |
|---|---|---|---|---|
|  | Time | Space | Time | Space |
| AdamW (fixed stepsize) | 0% | 0% | 0% | 0% |
| DoG (Ivgi et al., 2023) | +45% | 1.4% | +268% | 0% |
| gdtuo (Chandra et al., 2022) | +85% | 64% | +150% | 21% |
| mechanic (Cutkosky et al., 2024) | +42% | 88% | +9% | 0% |
| Prodigy (Mishchenko & Defazio, 2023) | +42% | 13% | +9% | 0% |
| MetaOptimize (AdamW, Lion) | +44% | 33% | +13% | 0% |

## 7.3 LANGUAGE MODELING

For language model experiments, we used the TinyStories dataset (Eldan & Li, 2023), a synthetic collection of brief stories designed for children aged 3 to 4. This dataset proves effective for training and evaluating language models that are significantly smaller than the current state-of-the-art, and capable of crafting stories that are not only fluent and coherent but also diverse.

We used the implementation in (Karpathy, 2024) for training 15M parameter model with a batch size of 128 on the TinyStories dataset. Two combinations of Hessian-free MetaOptimize with scalar

step sizes were tested against Lion and AdamW with well-tuned fixed step sizes, AdamW with a well-tuned cosine decay learning rate scheduler with 1k warmup iterations, and the four state-of-the-art step-size adaptation algorithms mentioned in the previous subsection. According to the learning curves, shown in Fig. 5, MetaOptimize outperforms all baselines (with an initial delay due to small initial step-sizes), except for the well-tuned learning rate scheduler within 30k iterations.

### 7.4 SENSITIVITY ANALYSIS

Here, we briefly discussion the sensitivity of MetaOptimize to its meta-meta-parameters.

For the meta-stepsize $\eta$ in MetaOptimize, there is generally no need for tuning, and the default value $\eta = 10^{-3}$ works universally well in stationary supervised learning. All experiments in this section used this default value with no sweeping required. The rationale for this choice is that when using Adam, Lion, or RMSProp for meta-updates, the absolute change in $\beta$ per iteration is approximately $\eta \times O(1) \simeq 10^{-3}$. Unless the current stepsize $\alpha$ is already near its optimal value, most $\beta$ updates will consistently move toward the optimal $\beta$. Within 1,000 steps, $\beta$ can change by $O(1)$, nearly doubling or halving $\alpha = \exp(\beta)$. Over 10,000 iterations, $\alpha$ can adjust to stepsizes that are $e^{10} > 20,000$ times larger or smaller, allowing $\eta \simeq 10^{-3}$ to efficiently track optimal stepsizes while minimizing unnecessary fluctuations in $\alpha$.

Regarding the discount factor $\gamma$, we used the default value $\gamma = 1$ in all experiments and observed minimal sensitivity to $\gamma$ for values $\gamma \geq 0.999$ in a series of preliminary tests. However, performance begins to degrade with smaller values of $\gamma$.

## 8 RELATED WORKS

Automatic adaptation of step sizes, has been an important research topic in the literature of stochastic optimization. Several works aimed to remove the manual tuning of learning rates via adaptations of classical line search (Rolinek & Martius, 2018; Vaswani et al., 2019; Paquette & Scheinberg, 2020; Kunstner et al., 2023) and Polyak step size (Berrada et al., 2020; Loizou et al., 2021), stochastic proximal methods (Asi & Duchi, 2019), stochastic quadratic approximation (Schaul et al., 2013), hyper-gradient descent (Baydin et al., 2017), nested hyper-gradient descent (Chandra et al., 2022), distance to a solution adaptation (Ivgi et al., 2023; Defazio & Mishchenko, 2023; Mishchenko & Defazio, 2023), and online convex learning (Cutkosky et al., 2024). A limitation of most of these methods is their potential underperformance when their meta-parameters are not optimally configured for specific problems (Ivgi et al., 2023). Moreover, the primary focus of most of these methods is on minimizing immediate loss rather than considering the long-term effects of step sizes on future loss.

Normalization techniques proposed over past few years, such as AdaGrad (Duchi et al., 2011), RMSProp, and Adam have significantly enhanced the training process. While these algorithms show promise in the stationary problems, these normalization techniques do not optimize effective step sizes and are prone to have sub-optimal performance especially in the continual learning settings (Degris et al., 2024).

An early practical step-size optimization method was the Incremental-Delta-Bar-Delta (IDBD) algorithm, introduced in (Sutton, 1992), which aimed to optimize the step-size vector to minimize a specific form of quadratic loss functions in a continual setting. This algorithm was later extended for neural networks in (Xu et al., 2018; Donini et al., 2019), and further adapted in (Mahmood et al., 2012; Javed, 2020; Micaelli & Storkey, 2021) for different meta or base updates beyond SGD. However, the development of IDBD and its extensions included some implicit assumptions, notably overlooking the impact of step-size dynamics on the formulation of step-size update rules. These extensions are, in essence, special cases of the L-approximation within the MetaOptimize framework. The current work extends the IDBD research, significantly broadening the framework and establishing a solid basis for the derivations. IDBD and its extensions have been used in various machine learning tasks including independent component analysis (Schraudolph & Giannakopoulos, 1999), human motion tracking (Kehl & Van Gool, 2006), classification (Koop, 2007; Andrychowicz et al., 2016), and reinforcement learning (Xu et al., 2018; Young et al., 2018; Javed et al., 2024). Refer to (Sutton, 2022) for a comprehensive history of step-size optimization.

A related line of work is gradient-based bilevel optimization, initially introduced by Bengio (2000) and later expanded in (Maclaurin et al., 2015; Pedregosa, 2016; Franceschi et al., 2018; Gao et al., 2022). Recent advances, such as (Lorraine et al., 2020), enable the optimization of millions of hyperparameters. While bilevel optimization focuses on tuning hyperparameters to minimize validation loss through repeated full training runs of the base algorithm, MetaOptimize diverges significantly. Designed for continual learning, MetaOptimize optimizes meta-parameters on-the-fly during a single streaming run, without relying on validation loss. Instead, it minimizes online loss (or regret) directly, aligning with the continual learning framework where no validation or test sets exist, and data arrives sequentially.

There is also a line of research on the so-called parameter-free optimization that aims to remove the need for step-size tuning with almost no knowledge of the problem properties. Most of these methods are primarily designed for stochastic convex optimization (Luo & Schapire, 2015; Orabona & Pál, 2016), while more recent ones (Orabona & Tommasi, 2017; Ivgi et al., 2023) were applied to supervised learning tasks with small or moderate sample sizes.

## 9 LIMITATIONS AND FUTURE WORKS

Our work represents a step toward unlocking the potential of meta-parameter optimization, with substantial room for further exploration, some of which we outline here:

**Hessian:** We confined our experiments to Hessian-free methods for practicality, though Hessian-based algorithms could offer superior performance. These methods, however, face challenges requiring additional research. The Hessian matrix is notably noisy, impacting $\mathcal{H}_{t+1}$ multiplicatively, necessitating smoothing and clipping techniques. Additionally, the Hessian approximates the loss landscape's curvature but fails to account for non-differentiable curvatures, such as those from ReLU unit breakpoints, significant at training's end. From a computational perspective, developing low-complexity methods for approximate Hessian matrix products, especially for adjusting step-sizes at the layer and weight levels, is essential.

**More accurate traces:** As discussed in Section 3, accuracy of the backward approximation (5) may degrade for larger values of the meta-stepsize $\eta$. Eligibility traces in RL suffer from a similar problem, to resolve which more-sophisticated traces (e.g., Dutch traces) have been developed (see Chapter 11 of (Sutton & Barto, 2018)). Developing more accurate backward approximations for meta-parameter optimization can result in considerable improvements in performance and stability.

**Blockwise step-sizes:** While step sizes can vary much in granularity, our experiments focused on scalar and blockwise step-sizes. While increasing the number of step sizes is anticipated to enhance performance, our experimental findings in Section 7 reveal that this improvement is not consistent across the MetaOptimize approximations evaluated. Further investigation is needed in future research.

**Other approximations:** We explored a limited set of MetaOptimize's possible approximations, leaving a comprehensive analysis of various approximations for future research.

**Other meta-parameters:** Our study was limited to differentiable meta-parameters, not covering discrete ones like batch size or network layer count. We also did not investigate several significant differentiable meta-parameters beyond step-sizes, deferring such exploration to future work.

**Automatic Differentiation:** While certain versions of MetaOptimize, such as the L-Approximation, could be implemented using standard automatic differentiation software, its applicability to the general case of MetaOptimize remains unclear. Unlike updates for $w$ and $\beta$ (base and meta parameters), the $H$ matrix lacks an explicit incremental formula that can be easily handled by automatic differentiation. For some versions of MetaOptimize, including the Hessian-free approximations used in our experiments, automatic differentiation is unnecessary, as meta updates do not require additional differentiation. Exploring the scope and applicability of automatic differentiation across different MetaOptimize instances is an interesting direction for future research.

**Continual learning:** Although continual step-size optimization is primarily aimed at continual learning, this study focused on the stationary case, demonstrating MetaOptimize's competitiveness in a context that is particularly challenging for it. Investigating the framework within continual learning presents a promising direction for future research.

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

# Appendices

## A  STEP-SIZE OPTIMIZATION FOR DIFFERENT CHOICES OF BASE AND META UPDATES

In this appendix, we derive $G_t$ defined in (10) for different choices of algorithms for base and meta updates, and propose corresponding step-size optimization algorithms.

Consider the following partitions of $G_t$,

$$G_t^{\text{meta}} \overset{\text{def}}{=} \left[ \begin{array}{ccc} \dfrac{\mathrm{d}\,\boldsymbol{y}_{t+1}}{\mathrm{d}\,\boldsymbol{y}_t} & \dfrac{\mathrm{d}\,\boldsymbol{y}_{t+1}}{\mathrm{d}\,\boldsymbol{x}_t} & \dfrac{\mathrm{d}\,\boldsymbol{y}_{t+1}}{\mathrm{d}\,\boldsymbol{h}_t} \end{array} \right], \tag{18}$$

$$G_t^{\text{base}} \overset{\text{def}}{=} \left[ \begin{array}{ccc} \dfrac{\mathrm{d}\,\boldsymbol{x}_{t+1}}{\mathrm{d}\,\boldsymbol{y}_t} & \dfrac{\mathrm{d}\,\boldsymbol{x}_{t+1}}{\mathrm{d}\,\boldsymbol{x}_t} & \dfrac{\mathrm{d}\,\boldsymbol{x}_{t+1}}{\mathrm{d}\,\boldsymbol{h}_t} \\[8pt] \dfrac{\mathrm{d}\,\boldsymbol{h}_{t+1}}{\mathrm{d}\,\boldsymbol{y}_t} & \dfrac{\mathrm{d}\,\boldsymbol{h}_{t+1}}{\mathrm{d}\,\boldsymbol{x}_t} & \dfrac{\mathrm{d}\,\boldsymbol{h}_{t+1}}{\mathrm{d}\,\boldsymbol{h}_t} \end{array} \right]. \tag{19}$$

Then,

$$G_t = \left[ \begin{array}{ccc} \dfrac{\mathrm{d}\,\boldsymbol{y}_{t+1}}{\mathrm{d}\,\boldsymbol{y}_t} & \dfrac{\mathrm{d}\,\boldsymbol{y}_{t+1}}{\mathrm{d}\,\boldsymbol{x}_t} & \dfrac{\mathrm{d}\,\boldsymbol{y}_{t+1}}{\mathrm{d}\,\boldsymbol{h}_t} \\[8pt] \dfrac{\mathrm{d}\,\boldsymbol{x}_{t+1}}{\mathrm{d}\,\boldsymbol{y}_t} & \dfrac{\mathrm{d}\,\boldsymbol{x}_{t+1}}{\mathrm{d}\,\boldsymbol{x}_t} & \dfrac{\mathrm{d}\,\boldsymbol{x}_{t+1}}{\mathrm{d}\,\boldsymbol{h}_t} \\[8pt] \dfrac{\mathrm{d}\,\boldsymbol{h}_{t+1}}{\mathrm{d}\,\boldsymbol{y}_t} & \dfrac{\mathrm{d}\,\boldsymbol{h}_{t+1}}{\mathrm{d}\,\boldsymbol{x}_t} & \dfrac{\mathrm{d}\,\boldsymbol{h}_{t+1}}{\mathrm{d}\,\boldsymbol{h}_t} \end{array} \right] = \left[ \begin{array}{c} G_t^{\text{meta}} \\[4pt] \hline \\[-8pt] G_t^{\text{base}} \end{array} \right]. \tag{20}$$

In the sequel, we study base and meta updates separately, because $\text{Alg}_{\text{base}}$ and $\text{Alg}_{\text{meta}}$ impact disjoint sets of blocks in $G_t$. In particular, as we will see, the choice of $\text{Alg}_{\text{base}}$ only affects $G^{\text{base}}$ while the choice of $\text{Alg}_{\text{meta}}$ only affects $G^{\text{meta}}$.

**Notation conventions in all Appendices:** For any vector $\boldsymbol{v}$, we denote by $[\boldsymbol{v}]$ a diagonal matrix with diagonal entries derived from $\boldsymbol{v}$. We denote by $\sigma'(\boldsymbol{\beta}_t)$ the Jacobian of $\boldsymbol{\alpha}_t$ with respect to $\boldsymbol{\beta}_t$.

Before delving into computing $G_t^{\text{base}}$ and $G_t^{\text{meta}}$ for different base and meta algorithms, we further simplify these matrices.

### A.1  DERIVATION OF $G^{\text{META}}$ FOR DIFFERENT META UPDATES

We start by simplifying $G^{\text{meta}}$, and introducing some notations.

Note that the meta update has no dependence on internal variables, $\tilde{\boldsymbol{x}}$, of the base algorithm. As a result,

$$\frac{\mathrm{d}\,\boldsymbol{y}_{t+1}}{\mathrm{d}\,\tilde{\boldsymbol{x}}_t} = 0. \tag{21}$$

Then,

$$G_t^{\text{meta}} = \left[ \begin{array}{ccc} \frac{\mathrm{d}\,\boldsymbol{y}_{t+1}}{\mathrm{d}\,\boldsymbol{y}_t} & \frac{\mathrm{d}\,\boldsymbol{y}_{t+1}}{\mathrm{d}\,\boldsymbol{x}_t} & \frac{\mathrm{d}\,\boldsymbol{y}_{t+1}}{\mathrm{d}\,\boldsymbol{h}_t} \end{array} \right] = \left[ \begin{array}{cccc} \frac{\mathrm{d}\,\boldsymbol{y}_{t+1}}{\mathrm{d}\,\boldsymbol{y}_t} & \frac{\mathrm{d}\,\boldsymbol{y}_{t+1}}{\mathrm{d}\,\boldsymbol{w}_t} & \frac{\mathrm{d}\,\boldsymbol{y}_{t+1}}{\mathrm{d}\,\tilde{\boldsymbol{x}}_t} & \frac{\mathrm{d}\,\boldsymbol{y}_{t+1}}{\mathrm{d}\,\boldsymbol{h}_t} \end{array} \right] = \left[ \begin{array}{cccc} \frac{\mathrm{d}\,\boldsymbol{y}_{t+1}}{\mathrm{d}\,\boldsymbol{y}_t} & \frac{\mathrm{d}\,\boldsymbol{y}_{t+1}}{\mathrm{d}\,\boldsymbol{w}_t} & 0 & \frac{\mathrm{d}\,\boldsymbol{y}_{t+1}}{\mathrm{d}\,\boldsymbol{h}_t} \end{array} \right], \tag{22}$$

where the third equality is due to (21). Let

$$L_t \overset{\text{def}}{=} \left[ \begin{array}{c|c|c|c} \nabla f_t(\boldsymbol{w}_t)^T & 0 & 0 & 0 \\ \hline 0 & \nabla f_t(\boldsymbol{w}_t)^T & 0 & 0 \\ \hline 0 & 0 & \ddots & 0 \\ \hline 0 & 0 & 0 & \nabla f_t(\boldsymbol{w}_t)^T \end{array} \right] \begin{array}{l} \leftarrow 1 \\ \leftarrow 2 \\ \vdots \\ \leftarrow m \end{array} \tag{23}$$

and recall that $\boldsymbol{h}_t$ is a vectorization of $\mathcal{H}_t$. Then,

$$\mathcal{H}_t \nabla f_t(\boldsymbol{w}_t) = L_t \boldsymbol{h}_t. \tag{24}$$

We now proceed to derivation of $G^{\text{meta}}$ for different choices of $\text{Alg}_{\text{meta}}$.

### A.1.1 Meta SGD

Here, we consider SGD for the meta update (9),

$$\boldsymbol{\beta}_{t+1} = \boldsymbol{\beta}_t - \eta \widehat{\nabla_{\boldsymbol{\beta}} F}_t = \boldsymbol{\beta}_t - \eta \mathcal{H}_t^T \nabla f_t(\boldsymbol{w}_t), \tag{25}$$

where $\eta$ is a scalar, called the *meta step size*. In this case, $\boldsymbol{y}_t = \boldsymbol{\beta}_t$. It then follows from (25) that

$$\frac{\mathrm{d}\boldsymbol{\beta}_{t+1}}{\mathrm{d}\boldsymbol{h}_t} = -\eta \frac{\mathrm{d}}{\mathrm{d}\boldsymbol{h}_t}\left(\mathcal{H}_t^T \nabla f_t(\boldsymbol{w}_t)\right) = -\eta \frac{\mathrm{d}}{\mathrm{d}\boldsymbol{h}_t}\left(L_t \boldsymbol{h}_t\right) = -\eta L_t, \tag{26}$$

where the second equality is due to (24). Consequently, from (22), we obtain

$$\begin{aligned} G_t^{\text{meta}} &= \left[\begin{array}{cccc} \frac{\mathrm{d}\boldsymbol{y}_{t+1}}{\mathrm{d}\boldsymbol{y}_t} & \frac{\mathrm{d}\boldsymbol{y}_{t+1}}{\mathrm{d}\boldsymbol{w}_t} & 0 & \frac{\mathrm{d}\boldsymbol{y}_{t+1}}{\mathrm{d}\boldsymbol{h}_t} \end{array}\right] \\ &= \left[\begin{array}{cccc} \frac{\mathrm{d}\boldsymbol{\beta}_{t+1}}{\mathrm{d}\boldsymbol{\beta}_t} & \frac{\mathrm{d}\boldsymbol{\beta}_{t+1}}{\mathrm{d}\boldsymbol{w}_t} & 0 & \frac{\mathrm{d}\boldsymbol{\beta}_{t+1}}{\mathrm{d}\boldsymbol{h}_t} \end{array}\right] \\ &= \left[\begin{array}{cccc} I & -\eta \mathcal{H}_t^T \nabla^2 f_t(\boldsymbol{w}_t) & 0 & -\eta L_t \end{array}\right], \end{aligned} \tag{27}$$

where the last inequality follows from (26) and simple differentiations of (25). Here, $\nabla^2 f_t(\boldsymbol{w}_t)$ denotes the Hessian of $f_t$ at $\boldsymbol{w}_t$.

### A.1.2 Meta Adam

The meta update based on the Adam algorithm is as follows,

$$\begin{aligned} \bar{\boldsymbol{m}}_{t+1} &= \bar{\rho}\,\bar{\boldsymbol{m}}_t + \mathcal{H}_t^T \nabla f_t(\boldsymbol{w}_t), \\ \bar{\boldsymbol{v}}_{t+1} &= \bar{\lambda}\,\boldsymbol{v}_t + \left(\mathcal{H}_t^T \nabla f_t(\boldsymbol{w}_t)\right)^2, \\ \bar{\mu}_t &= \left(\frac{1-\bar{\rho}}{1-\bar{\rho}^t}\right) \Big/ \sqrt{\frac{1-\bar{\lambda}}{1-\bar{\lambda}^t}}, \\ \boldsymbol{\beta}_{t+1} &= \boldsymbol{\beta}_t - \eta\,\bar{\mu}_t \frac{\bar{\boldsymbol{m}}_t}{\sqrt{\bar{\boldsymbol{v}}_t}} \end{aligned} \tag{28}$$

where $\bar{\boldsymbol{m}}_t$ is the momentum vector, $\bar{\boldsymbol{v}}_t$ is the trace of squared surrogate-meta-gradient. Since Adam algorithm needs to keep track of $\boldsymbol{\beta}_t$, $\bar{\boldsymbol{m}}_t$, and $\bar{\boldsymbol{v}}_t$, we have

$$\boldsymbol{y}_t = \left[\begin{array}{c} \boldsymbol{\beta}_t \\ \bar{\boldsymbol{m}}_t \\ \bar{\boldsymbol{v}}_t \end{array}\right]. \tag{29}$$

Recall the following notation convention at the end of the Introduction section: for any $k \geq 1$, and any $k$-dimensional vector $\boldsymbol{v} = [v_1, \ldots, v_k]$, we denote the the corresponding diagonal matrix by $[\boldsymbol{v}]$:

$$[\boldsymbol{v}] \stackrel{\text{def}}{=} \left[\begin{array}{ccc} v_1 & \cdots & 0 \\ \vdots & \ddots & \vdots \\ 0 & \cdots & v_k \end{array}\right]. \tag{30}$$

Consequently, from (22), we obtain

$$\begin{aligned} G_t^{\text{meta}} &= \left[\begin{array}{cc|c|c} \frac{\mathrm{d}\boldsymbol{y}_{t+1}}{\mathrm{d}\boldsymbol{y}_t} & \frac{\mathrm{d}\boldsymbol{y}_{t+1}}{\mathrm{d}\boldsymbol{w}_t} & 0 & \frac{\mathrm{d}\boldsymbol{y}_{t+1}}{\mathrm{d}\boldsymbol{h}_t} \end{array}\right] \\ &= \left[\begin{array}{ccc|cc|c} \frac{\mathrm{d}\boldsymbol{\beta}_{t+1}}{\mathrm{d}\boldsymbol{\beta}_t} & \frac{\mathrm{d}\boldsymbol{\beta}_{t+1}}{\mathrm{d}\bar{\boldsymbol{m}}_t} & \frac{\mathrm{d}\boldsymbol{\beta}_{t+1}}{\mathrm{d}\bar{\boldsymbol{v}}_t} & \frac{\mathrm{d}\boldsymbol{\beta}_{t+1}}{\mathrm{d}\boldsymbol{w}_t} & 0 & \frac{\mathrm{d}\boldsymbol{\beta}_{t+1}}{\mathrm{d}\boldsymbol{h}_t} \\ \frac{\mathrm{d}\bar{\boldsymbol{m}}_{t+1}}{\mathrm{d}\boldsymbol{\beta}_t} & \frac{\mathrm{d}\bar{\boldsymbol{m}}_{t+1}}{\mathrm{d}\bar{\boldsymbol{m}}_t} & \frac{\mathrm{d}\bar{\boldsymbol{m}}_{t+1}}{\mathrm{d}\bar{\boldsymbol{v}}_t} & \frac{\mathrm{d}\bar{\boldsymbol{m}}_{t+1}}{\mathrm{d}\boldsymbol{w}_t} & 0 & \frac{\mathrm{d}\bar{\boldsymbol{m}}_{t+1}}{\mathrm{d}\boldsymbol{h}_t} \\ \frac{\mathrm{d}\bar{\boldsymbol{v}}_{t+1}}{\mathrm{d}\boldsymbol{\beta}_t} & \frac{\mathrm{d}\bar{\boldsymbol{v}}_{t+1}}{\mathrm{d}\bar{\boldsymbol{m}}_t} & \frac{\mathrm{d}\bar{\boldsymbol{v}}_{t+1}}{\mathrm{d}\bar{\boldsymbol{v}}_t} & \frac{\mathrm{d}\bar{\boldsymbol{v}}_{t+1}}{\mathrm{d}\boldsymbol{w}_t} & 0 & \frac{\mathrm{d}\bar{\boldsymbol{v}}_{t+1}}{\mathrm{d}\boldsymbol{h}_t} \end{array}\right] \\ &= \left[\begin{array}{ccc|cc|c} I & -\eta\bar{\mu}_t\left[\frac{1}{\sqrt{\bar{\boldsymbol{v}}_t}}\right] & \frac{\eta\bar{\mu}_t}{2}\left[\frac{\bar{\boldsymbol{m}}_t}{\bar{\boldsymbol{v}}_t^{1.5}}\right] & 0 & 0 & 0 \\ 0 & \bar{\rho}I & 0 & \mathcal{H}_t^T \nabla^2 f_t & 0 & \frac{\mathrm{d}\bar{\boldsymbol{m}}_{t+1}}{\mathrm{d}\boldsymbol{h}_t} \\ 0 & 0 & \bar{\lambda}I & 2\left[\mathcal{H}_t^T \nabla f_t\right]\mathcal{H}_t^T \nabla^2 f_t & 0 & \frac{\mathrm{d}\bar{\boldsymbol{v}}_{t+1}}{\mathrm{d}\boldsymbol{h}_t} \end{array}\right], \end{aligned} \tag{31}$$

where the last equality follows by calculating derivatives of (28). For the two remaining terms in the last column of $G_t$, we have

$$\frac{\mathrm{d}\,\bar{\boldsymbol{m}}_{t+1}}{\mathrm{d}\,\boldsymbol{h}_t} \;=\; \frac{\mathrm{d}}{\mathrm{d}\,\boldsymbol{h}_t}\big(\mathcal{H}_t^T \nabla f_t(\boldsymbol{w}_t)\big) \;=\; \eta\frac{\mathrm{d}}{\mathrm{d}\,\boldsymbol{h}_t}\big(L_t\boldsymbol{h}_t\big) \;=\; \eta\,L_t. \tag{32}$$

where the first equality follows from the update of $\bar{\boldsymbol{m}}_{t+1}$ in (28), and the second equality is due to (24). In the same vein,

$$\frac{\mathrm{d}\,\bar{\boldsymbol{v}}_{t+1}}{\mathrm{d}\,\boldsymbol{h}_t} \;=\; \frac{\mathrm{d}}{\mathrm{d}\,\boldsymbol{h}_t}\big(\mathcal{H}_t^T \nabla f_t(\boldsymbol{w}_t)\big)^2 \;=\; \frac{\mathrm{d}}{\mathrm{d}\,\boldsymbol{h}_t}\big(L_t\boldsymbol{h}_t\big)^2 \;=\; 2\big[L_t\boldsymbol{h}_t\big]\frac{\mathrm{d}}{\mathrm{d}\,\boldsymbol{h}_t}\big(L_t\boldsymbol{h}_t\big) \;=\; 2\big[L_t\boldsymbol{h}_t\big]\,L_t \;=\; 2\big[\mathcal{H}_t^T \nabla f_t(\boldsymbol{w}_t)\big]\,L_t, \tag{33}$$

where the first equality follows from the update of $\bar{\boldsymbol{v}}_{t+1}$ in (28), the second equality is due to (24), and the last equality is again from (24).

Plugging (32) and (33) into (31), we obtain

$$G_t^{\mathrm{meta}} = \left[\begin{array}{ccc|ccc}
I & -\eta\bar{\mu}_t\left[\frac{1}{\sqrt{\bar{\boldsymbol{v}}_t}}\right] & \frac{\eta\bar{\mu}_t}{2}\left[\frac{\bar{\boldsymbol{m}}_t}{\bar{\boldsymbol{v}}_t^{1.5}}\right] & 0 & 0 & 0 \\
0 & \bar{\rho}I & 0 & \mathcal{H}_t^T\nabla^2 f_t & 0 & \eta\,L_t \\
0 & 0 & \bar{\lambda}I & 2\big[\mathcal{H}_t^T\nabla f_t\big]\mathcal{H}_t^T\nabla^2 f_t & 0 & 2\big[\mathcal{H}_t^T\nabla f_t\big]\,L_t
\end{array}\right]. \tag{34}$$

### A.1.3  Meta Lion

The meta update based on the lion algorithm is as follows

$$\bar{\boldsymbol{m}}_{t+1} = \rho\,\bar{\boldsymbol{m}}_t + (1-\rho)\,\widehat{\nabla_{\boldsymbol{\beta}}F}_t, \tag{35}$$

$$\boldsymbol{\beta}_{t+1} = \boldsymbol{\beta}_t - \eta\,\mathrm{Sign}\big(c\,\bar{\boldsymbol{m}}_t + (1-c)\widehat{\nabla_{\boldsymbol{\beta}}F}_t\big), \tag{36}$$

where $\eta$ is a scalar, called the *meta step size*, and $\rho, c \in [0,1)$. Note that the meta algorithm operates on a low dimensional space. Therefore, we drop the regularizers like weight-decay in the meta updates, as they are primarily aimed to resolve the overfitting problem in high dimensional problems. Substituting $\widehat{\nabla_{\boldsymbol{\beta}}F}_t$ with $\mathcal{H}_t^T\nabla f_t(\boldsymbol{w}_t)$ we obtain the following meta updates

$$\bar{\boldsymbol{m}}_{t+1} = \rho\,\bar{\boldsymbol{m}}_t + (1-\rho)\,\mathcal{H}_t^T\nabla f_t(\boldsymbol{w}_t), \tag{37}$$

$$\boldsymbol{\beta}_{t+1} = \boldsymbol{\beta}_t - \eta\,\mathrm{Sign}\big(c\,\bar{\boldsymbol{m}}_t + (1-c)\mathcal{H}_t^T\nabla f_t(\boldsymbol{w}_t)\big). \tag{38}$$

In this case,

$$\boldsymbol{y}_t = \left[\begin{array}{c} \boldsymbol{\beta}_t \\ \bar{\boldsymbol{m}}_t \end{array}\right],$$

and

$$\begin{aligned}
G_t^{\mathrm{meta}} &= \left[\begin{array}{cccc} \frac{\mathrm{d}\,\boldsymbol{y}_{t+1}}{\mathrm{d}\,\boldsymbol{y}_t} & \frac{\mathrm{d}\,\boldsymbol{y}_{t+1}}{\mathrm{d}\,\boldsymbol{w}_t} & 0 & \frac{\mathrm{d}\,\boldsymbol{y}_{t+1}}{\mathrm{d}\,\boldsymbol{h}_t} \end{array}\right] \\
&= \left[\begin{array}{ccccc} \frac{\mathrm{d}\,\boldsymbol{\beta}_{t+1}}{\mathrm{d}\,\boldsymbol{\beta}_t} & \frac{\mathrm{d}\,\boldsymbol{\beta}_{t+1}}{\mathrm{d}\,\bar{\boldsymbol{m}}_t} & \frac{\mathrm{d}\,\boldsymbol{\beta}_{t+1}}{\mathrm{d}\,\boldsymbol{w}_t} & 0 & \frac{\mathrm{d}\,\boldsymbol{\beta}_{t+1}}{\mathrm{d}\,\boldsymbol{h}_t} \\[4pt] \frac{\mathrm{d}\,\bar{\boldsymbol{m}}_{t+1}}{\mathrm{d}\,\boldsymbol{\beta}_t} & \frac{\mathrm{d}\,\bar{\boldsymbol{m}}_{t+1}}{\mathrm{d}\,\bar{\boldsymbol{m}}_t} & \frac{\mathrm{d}\,\bar{\boldsymbol{m}}_{t+1}}{\mathrm{d}\,\boldsymbol{w}_t} & 0 & \frac{\mathrm{d}\,\bar{\boldsymbol{m}}_{t+1}}{\mathrm{d}\,\boldsymbol{h}_t} \end{array}\right] \\
&= \left[\begin{array}{ccccc} I & 0 & 0 & 0 & 0 \\[4pt] \frac{\mathrm{d}\,\bar{\boldsymbol{m}}_{t+1}}{\mathrm{d}\,\boldsymbol{\beta}_t} & \frac{\mathrm{d}\,\bar{\boldsymbol{m}}_{t+1}}{\mathrm{d}\,\bar{\boldsymbol{m}}_t} & \frac{\mathrm{d}\,\bar{\boldsymbol{m}}_{t+1}}{\mathrm{d}\,\boldsymbol{w}_t} & 0 & \frac{\mathrm{d}\,\bar{\boldsymbol{m}}_{t+1}}{\mathrm{d}\,\boldsymbol{h}_t} \end{array}\right],
\end{aligned} \tag{39}$$

where the last equality follows from (38). Consider the following block representation of $Y_t$:

$$Y_t = \left[\begin{array}{c} B_t \\ Y_t^{\bar{m}} \end{array}\right]. \tag{40}$$

Since the base algorithm, does not take $\bar{\boldsymbol{m}}$ as input, as we will see in (42) and (43) of next subsection (Appendix A.2), $\frac{\mathrm{d}\,\bar{\boldsymbol{m}}_{t+1}}{\mathrm{d}\,\bar{\boldsymbol{m}}_t}$ is the only non-zero block of $G_t$ in its column of blocks (i.e., $\frac{\mathrm{d}\,s_{t+1}}{\mathrm{d}\,\bar{\boldsymbol{m}}_t} = 0$ for every variable $s$ other than $\bar{\boldsymbol{m}}$). Consequently, it follows from (15) that $Y_t^{\bar{m}}$ as defined in (40), has no impact on the update of $X_{t+1}$, $B_{t+1}$, and $Q_{t+1}$. Therefore, we can zero-out the rows and columns of $G^{\mathrm{meta}}$ that correspond to derivative of $\bar{\boldsymbol{m}}$. As such we obtain the following equivalent of $G^{\mathrm{meta}}$ in (39) from an algorithmic perspective:

$$G_t^{\mathrm{meta}} \equiv \left[\begin{array}{cc} I_{m\times m} & 0 \\ 0 & 0 \end{array}\right]. \tag{41}$$

As a result, we get $B_t = I$ for all times $t$.

## A.2 Derivation of $G^{\text{base}}$ for Different Base Updates

We now turn our focus to computation of $G^{\text{base}}$. Let us start by simplifying $G^{\text{base}}$, and introducing some notations.

Note that the base update has no dependence on internal variables, $\tilde{\boldsymbol{y}}$, of the meta update. As a result,

$$\frac{\mathrm{d}\,\boldsymbol{x}_{t+1}}{\mathrm{d}\,\tilde{\boldsymbol{y}}_t} = 0. \tag{42}$$

Moreover, it follows from the definition of $\mathcal{H}_t$ in (8) that

$$\frac{\mathrm{d}\,\mathcal{H}_{t+1}}{\mathrm{d}\,\tilde{\boldsymbol{y}}_t} = (1-\gamma)\sum_{t=0}^{t}\gamma^{t-\tau}\frac{\mathrm{d}}{d\tilde{\boldsymbol{y}}_t}\left(\frac{d\boldsymbol{w}_{t+1}}{\mathrm{d}\,\boldsymbol{\beta}_\tau}\right) = (1-\gamma)\sum_{t=0}^{t}\gamma^{t-\tau}\frac{\mathrm{d}}{d\boldsymbol{\beta}_\tau}\left(\frac{d\boldsymbol{w}_{t+1}}{\mathrm{d}\,\tilde{\boldsymbol{y}}_t}\right) = (1-\gamma)\sum_{t=0}^{t}\gamma^{t-\tau}\frac{\mathrm{d}}{d\boldsymbol{\beta}_\tau}\,(0) = 0,$$

where the third equality follows from (42). Therefore,

$$\frac{\mathrm{d}\,\boldsymbol{h}_{t+1}}{\mathrm{d}\,\tilde{\boldsymbol{y}}_t} = 0. \tag{43}$$

Note also that $\mathrm{Alg}_{\text{base}}$ does not take $\mathcal{H}_t$ as input, and therefore,

$$\frac{\mathrm{d}\,\boldsymbol{x}_{t+1}}{\mathrm{d}\,\boldsymbol{h}_t} = 0. \tag{44}$$

Consequently, we can simplify $G_t^{\text{base}}$ as follows,

$$G_t^{\text{base}} = \begin{bmatrix} \frac{\mathrm{d}\,\boldsymbol{x}_{t+1}}{\mathrm{d}\,\boldsymbol{y}_t} & \frac{\mathrm{d}\,\boldsymbol{x}_{t+1}}{\mathrm{d}\,\boldsymbol{x}_t} & \frac{\mathrm{d}\,\boldsymbol{x}_{t+1}}{\mathrm{d}\,\boldsymbol{h}_t} \\ \frac{\mathrm{d}\,\boldsymbol{h}_{t+1}}{\mathrm{d}\,\boldsymbol{y}_t} & \frac{\mathrm{d}\,\boldsymbol{h}_{t+1}}{\mathrm{d}\,\boldsymbol{x}_t} & \frac{\mathrm{d}\,\boldsymbol{h}_{t+1}}{\mathrm{d}\,\boldsymbol{h}_t} \end{bmatrix} = \begin{bmatrix} \frac{\mathrm{d}\,\boldsymbol{x}_{t+1}}{\mathrm{d}\,\boldsymbol{\beta}_t} & \frac{\mathrm{d}\,\boldsymbol{x}_{t+1}}{\mathrm{d}\,\tilde{\boldsymbol{y}}_t} & \frac{\mathrm{d}\,\boldsymbol{x}_{t+1}}{\mathrm{d}\,\boldsymbol{x}_t} & \frac{\mathrm{d}\,\boldsymbol{x}_{t+1}}{\mathrm{d}\,\boldsymbol{h}_t} \\ \frac{\mathrm{d}\,\boldsymbol{h}_{t+1}}{\mathrm{d}\,\boldsymbol{\beta}_t} & \frac{\mathrm{d}\,\boldsymbol{h}_{t+1}}{\mathrm{d}\,\tilde{\boldsymbol{y}}_t} & \frac{\mathrm{d}\,\boldsymbol{h}_{t+1}}{\mathrm{d}\,\boldsymbol{x}_t} & \frac{\mathrm{d}\,\boldsymbol{h}_{t+1}}{\mathrm{d}\,\boldsymbol{h}_t} \end{bmatrix} = \begin{bmatrix} \frac{\mathrm{d}\,\boldsymbol{x}_{t+1}}{\mathrm{d}\,\boldsymbol{\beta}_t} & 0 & \frac{\mathrm{d}\,\boldsymbol{x}_{t+1}}{\mathrm{d}\,\boldsymbol{x}_t} & 0 \\ \frac{\mathrm{d}\,\boldsymbol{h}_{t+1}}{\mathrm{d}\,\boldsymbol{\beta}_t} & 0 & \frac{\mathrm{d}\,\boldsymbol{h}_{t+1}}{\mathrm{d}\,\boldsymbol{x}_t} & \frac{\mathrm{d}\,\boldsymbol{h}_{t+1}}{\mathrm{d}\,\boldsymbol{h}_t} \end{bmatrix}, \tag{45}$$

where the last equality is due to (42), (43), and (44).

On an independent note, consider the following block representation of $Y_t$,

$$Y_t = \begin{bmatrix} B_t - \frac{1-\gamma}{\gamma}I \\ \tilde{Y}_t \end{bmatrix}, \tag{46}$$

Therefore,

$$\gamma Y_t + (1-\gamma)\begin{bmatrix} I \\ 0 \end{bmatrix} = \begin{bmatrix} B_t \\ \tilde{Y}_t \end{bmatrix}$$

It then follows from (20) and (15) that

$$\begin{bmatrix} X_{t+1} \\ Q_{t+1} \end{bmatrix} = \gamma\, G_t^{\text{base}}\begin{bmatrix} \begin{bmatrix} B_t \\ \tilde{Y}_t \end{bmatrix} \\ X_t \\ Q_t \end{bmatrix}. \tag{47}$$

Moreover, from the definition of $Y_t$ in (12), we have

$$\frac{\mathrm{d}\,B_t}{\mathrm{d}\,\boldsymbol{x}_t} = (1-\gamma)\frac{\mathrm{d}}{\mathrm{d}\,\boldsymbol{x}_t}\sum_{\tau=0}^{t}\gamma^{t-\tau}\frac{\mathrm{d}\,\boldsymbol{\beta}_t}{\mathrm{d}\,\boldsymbol{\beta}_\tau} = (1-\gamma)\sum_{\tau=0}^{t}\gamma^{t-\tau}\frac{\mathrm{d}}{\mathrm{d}\,\boldsymbol{\beta}_\tau}\left(\frac{\mathrm{d}\,\boldsymbol{\beta}_t}{\mathrm{d}\,\boldsymbol{x}_t}\right) = (1-\gamma)\sum_{\tau=0}^{t}\gamma^{t-\tau}\frac{\mathrm{d}}{\mathrm{d}\,\boldsymbol{\beta}_\tau}\,(0) = 0,$$

$$\frac{\mathrm{d}\,B_t}{\mathrm{d}\,\boldsymbol{\beta}_t} = (1-\gamma)\frac{\mathrm{d}}{\mathrm{d}\,\boldsymbol{\beta}_t}\sum_{\tau=0}^{t}\gamma^{t-\tau}\frac{\mathrm{d}\,\boldsymbol{\beta}_t}{\mathrm{d}\,\boldsymbol{\beta}_\tau} = (1-\gamma)\sum_{\tau=0}^{t}\gamma^{t-\tau}\frac{\mathrm{d}}{\mathrm{d}\,\boldsymbol{\beta}_\tau}\left(\frac{\mathrm{d}\,\boldsymbol{\beta}_t}{\mathrm{d}\,\boldsymbol{\beta}_t}\right) = (1-\gamma)\sum_{\tau=0}^{t}\gamma^{t-\tau}\frac{\mathrm{d}}{\mathrm{d}\,\boldsymbol{\beta}_\tau}\,(I) = 0,$$

$$\frac{\mathrm{d}\,B_t}{\mathrm{d}\,\boldsymbol{h}_t} = (1-\gamma)\frac{\mathrm{d}}{\mathrm{d}\,\boldsymbol{h}_t}\sum_{\tau=0}^{t}\gamma^{t-\tau}\frac{\mathrm{d}\,\boldsymbol{\beta}_t}{\mathrm{d}\,\boldsymbol{\beta}_\tau} = (1-\gamma)\sum_{\tau=0}^{t}\gamma^{t-\tau}\frac{\mathrm{d}}{\mathrm{d}\,\boldsymbol{\beta}_\tau}\left(\frac{\mathrm{d}\,\boldsymbol{\beta}_t}{\mathrm{d}\,\boldsymbol{h}_t}\right) = (1-\gamma)\sum_{\tau=0}^{t}\gamma^{t-\tau}\frac{\mathrm{d}}{\mathrm{d}\,\boldsymbol{\beta}_\tau}\,(0) = 0. \tag{48}$$

Finally, recall the definition

$$\sigma'(\boldsymbol{\beta}_t) \stackrel{\text{def}}{=} \frac{\mathrm{d}\,\boldsymbol{\alpha}_t}{\mathrm{d}\,\boldsymbol{\beta}_t} \tag{49}$$

as the Jacobian of $\boldsymbol{\alpha}_t$ with respect to $\boldsymbol{\beta}_t$.

We now proceed to derivation of $G^{\text{base}}$ for different choices of $\mathrm{Alg}_{\text{base}}$.

### A.3   BASE SGD

Base SGD algorithm makes the following base update in each iteration:

$$\boldsymbol{w}_{t+1} = \boldsymbol{w}_t - \boldsymbol{\alpha}_t \nabla f_t(\boldsymbol{w}_t). \tag{50}$$

In this case, $\boldsymbol{x}_t = \boldsymbol{w}_t$ and $X_t = \mathcal{H}_t$. Then, $G_t^{\text{base}}$ in (45) can be simplified to

$$
\begin{aligned}
G_t^{\text{base}} &= \begin{bmatrix} \frac{\mathrm{d}\,\boldsymbol{x}_{t+1}}{\mathrm{d}\,\boldsymbol{\beta}_t} & 0 & \frac{\mathrm{d}\,\boldsymbol{x}_{t+1}}{\mathrm{d}\,\boldsymbol{x}_t} & 0 \\ \frac{\mathrm{d}\,\boldsymbol{h}_{t+1}}{\mathrm{d}\,\boldsymbol{\beta}_t} & 0 & \frac{\mathrm{d}\,\boldsymbol{h}_{t+1}}{\mathrm{d}\,\boldsymbol{x}_t} & \frac{\mathrm{d}\,\boldsymbol{h}_{t+1}}{\mathrm{d}\,\boldsymbol{h}_t} \end{bmatrix} \\
&= \begin{bmatrix} \frac{\mathrm{d}\,\boldsymbol{w}_{t+1}}{\mathrm{d}\,\boldsymbol{\beta}_t} & 0 & \frac{\mathrm{d}\,\boldsymbol{w}_{t+1}}{\mathrm{d}\,\boldsymbol{w}_t} & 0 \\ \frac{\mathrm{d}\,\boldsymbol{h}_{t+1}}{\mathrm{d}\,\boldsymbol{\beta}_t} & 0 & \frac{\mathrm{d}\,\boldsymbol{h}_{t+1}}{\mathrm{d}\,\boldsymbol{w}_t} & \frac{\mathrm{d}\,\boldsymbol{h}_{t+1}}{\mathrm{d}\,\boldsymbol{h}_t} \end{bmatrix} \\
&= \begin{bmatrix} -\left[\nabla f_t(\boldsymbol{w}_t)\right]\sigma'(\boldsymbol{\beta}_t) & 0 & I - \left[\boldsymbol{\alpha}_t\right]\nabla^2 f_t(\boldsymbol{w}_t) & 0 \\ \frac{\mathrm{d}\,\boldsymbol{h}_{t+1}}{\mathrm{d}\,\boldsymbol{\beta}_t} & 0 & \frac{\mathrm{d}\,\boldsymbol{h}_{t+1}}{\mathrm{d}\,\boldsymbol{w}_t} & \frac{\mathrm{d}\,\boldsymbol{h}_{t+1}}{\mathrm{d}\,\boldsymbol{h}_t} \end{bmatrix},
\end{aligned}
\tag{51}
$$

where the last equality follows by computing simple derivatives of $\boldsymbol{w}_{t+1}$ in (50).

We proceed to compute the three remaining entries of $G_t^{\text{base}}$, i.e., $\mathrm{d}\,\boldsymbol{h}_{t+1}/\mathrm{d}\,\boldsymbol{\beta}_t$, $\mathrm{d}\,\boldsymbol{h}_{t+1}/\mathrm{d}\,\boldsymbol{w}_t$, and $\mathrm{d}\,\boldsymbol{h}_{t+1}/\mathrm{d}\,\boldsymbol{h}_t$. Note that by plugging the first row of $G_t^{\text{base}}$, given in (51), into (47), and noting that $\mathcal{H}_t = X_t$, we obtain

$$\mathcal{H}_{t+1} = \gamma\big(I - \left[\boldsymbol{\alpha}_t\right]\nabla^2 f_t(\boldsymbol{w}_t)\big)\mathcal{H}_t - \gamma\left[\nabla f_t(\boldsymbol{w}_t)\right]\sigma'(\boldsymbol{\beta}_t)\,B_t, \tag{52}$$

for all $t \geq 0$. By vectorizing both sides of (52) we obtain

$$\boldsymbol{h}_{t+1} = \gamma \begin{bmatrix} \left(I - \left[\boldsymbol{\alpha}_t\right]\nabla^2 f_t\right)\mathcal{H}_t^{[1]} - \left[\nabla f_t\right]\sigma'(\boldsymbol{\beta}_t)\,B_t^{[1]} \\ \left(I - \left[\boldsymbol{\alpha}_t\right]\nabla^2 f_t\right)\mathcal{H}_t^{[2]} - \left[\nabla f_t\right]\sigma'(\boldsymbol{\beta}_t)\,B_t^{[2]} \\ \vdots \\ \left(I - \left[\boldsymbol{\alpha}_t\right]\nabla^2 f_t\right)\mathcal{H}_t^{[m]} - \left[\nabla f_t\right]\sigma'(\boldsymbol{\beta}_t)\,B_t^{[m]} \end{bmatrix}. \tag{53}$$

Note that for any pair of same-size vectors $\boldsymbol{a}$ and $\boldsymbol{b}$, we have $[\boldsymbol{a}]\,\boldsymbol{b} = [\boldsymbol{b}]\,\boldsymbol{a}$ where $[\boldsymbol{a}]$ and $[\boldsymbol{b}]$ are diagonal matrices of $\boldsymbol{a}$ and $\boldsymbol{b}$, respectively. Therefore, (53) can be equivalently written in the following form

$$\boldsymbol{h}_{t+1} = \gamma \begin{bmatrix} \left(I - \left[\boldsymbol{\alpha}_t\right]\nabla^2 f_t\right)\mathcal{H}_t^{[1]} - \left[\sigma'(\boldsymbol{\beta}_t)\,B_t^{[1]}\right]\nabla f_t \\ \vdots \\ \left(I - \left[\boldsymbol{\alpha}_t\right]\nabla^2 f_t\right)\mathcal{H}_t^{[m]} - \left[\sigma'(\boldsymbol{\beta}_t)\,B_t^{[m]}\right]\nabla f_t \end{bmatrix}. \tag{54}$$

By taking the derivative of (53) with respect to $\boldsymbol{h}_t$, we obtain

$$\frac{\mathrm{d}\,\boldsymbol{h}_{t+1}}{\mathrm{d}\,\boldsymbol{h}_t} = \gamma \begin{bmatrix} I - \left[\boldsymbol{\alpha}_t\right]\nabla^2 f_t(\boldsymbol{w}_t) & 0 & 0 & 0 \\ 0 & I - \left[\boldsymbol{\alpha}_t\right]\nabla^2 f_t(\boldsymbol{w}_t) & 0 & 0 \\ 0 & 0 & \ddots & 0 \\ 0 & 0 & 0 & I - \left[\boldsymbol{\alpha}_t\right]\nabla^2 f_t(\boldsymbol{w}_t) \end{bmatrix} \begin{matrix} \leftarrow \text{1st} \\ \leftarrow \text{2nd} \\ \vdots \\ \leftarrow m\text{th} \end{matrix}. \tag{55}$$

In the above equation, note that $\mathrm{d}\,B_t/\mathrm{d}\,\boldsymbol{h}_t = 0$ due to (48). Let $\beta_t[i]$ and $w_t[j]$ denote the $i$th and $j$th entries of $\boldsymbol{\beta}_t$ and $\boldsymbol{w}_t$, for $i = 1,\ldots,m$ and $j = 1,\ldots,n$, respectively. It then follows from (53) and (48) that

$$\frac{\mathrm{d}\,\boldsymbol{h}_{t+1}}{\mathrm{d}\,\boldsymbol{\beta}_t} = -\gamma \begin{bmatrix} \left[\frac{\mathrm{d}\,\boldsymbol{\alpha}_t}{\mathrm{d}\,\beta_t[1]}\right]\nabla^2 f_t\,\mathcal{H}_t^{[1]} + \left[\nabla f_t\right]\frac{\partial\,\sigma'(\boldsymbol{\beta}_t)}{\partial\,\beta_t[1]}B_t^{[1]} & \cdots & \left[\frac{\mathrm{d}\,\boldsymbol{\alpha}_t}{\mathrm{d}\,\beta_t[m]}\right]\nabla^2 f_t\,\mathcal{H}_t^{[1]} + \left[\nabla f_t\right]\frac{\partial\,\sigma'(\boldsymbol{\beta}_t)}{\partial\,\beta_t[m]}B_t^{[1]} \\ \vdots & \ddots & \vdots \\ \left[\frac{\mathrm{d}\,\boldsymbol{\alpha}_t}{\mathrm{d}\,\beta_t[1]}\right]\nabla^2 f_t\,\mathcal{H}_t^{[m]} + \left[\nabla f_t\right]\frac{\partial\,\sigma'(\boldsymbol{\beta}_t)}{\partial\,\beta_t[1]}B_t^{[m]} & \cdots & \left[\frac{\mathrm{d}\,\boldsymbol{\alpha}_t}{\mathrm{d}\,\beta_t[m]}\right]\nabla^2 f_t\,\mathcal{H}_t^{[m]} + \left[\nabla f_t\right]\frac{\partial\,\sigma'(\boldsymbol{\beta}_t)}{\partial\,\beta_t[m]}B_t^{[m]} \end{bmatrix}, \tag{56}$$

where $\frac{\partial}{\partial \beta}$ stands for the entry-wise partial derivative of a matrix with respect to a scalar variable $\beta$. In the same vein, (54) and (48) imply that

$$
\frac{\mathrm{d}\,\boldsymbol{h}_{t+1}}{\mathrm{d}\,\boldsymbol{w}_t} = -\gamma \left[ \begin{array}{c} [\boldsymbol{\alpha}_t]\,\frac{\mathrm{d}\,(\nabla^2 f_t(\boldsymbol{w}_t)\,\mathcal{H}_t^{[1]})}{\mathrm{d}\,\boldsymbol{w}_t}\ +\ \left[\sigma'(\boldsymbol{\beta}_t)\,B_t^{[1]}\right]\nabla^2 f_t(\boldsymbol{w}_t) \\ \vdots \\ [\boldsymbol{\alpha}_t]\,\frac{\mathrm{d}\,(\nabla^2 f_t(\boldsymbol{w}_t)\,\mathcal{H}_t^{[m]})}{\mathrm{d}\,\boldsymbol{w}_t}\ +\ \left[\sigma'(\boldsymbol{\beta}_t)\,B_t^{[m]}\right]\nabla^2 f_t(\boldsymbol{w}_t) \end{array} \right]. \tag{57}
$$

Finally, $G_t^{\text{base}}$ is obtained by plugging (55), (56), and (57) into (51).

In the **special case that $\beta$ is a scalar** (equivalently $m = 1$), and furthermore $\alpha = \sigma(\beta) = e^\beta$, matrix $G_t^{\text{base}}$ would be simplified to

$$
G_t^{\text{base (scalar)}} = \left[ \begin{array}{ccc} 1 & -\eta\,\boldsymbol{h}_t^T \nabla^2 f_t(\boldsymbol{w}_t) & -\eta\,\nabla f_t(\boldsymbol{w}_t)^T \\ -\alpha \nabla f_t(\boldsymbol{w}_t) & I - \alpha \nabla^2 f_t(\boldsymbol{w}_t) & 0 \\ -\gamma\alpha\nabla^2 f_t(\boldsymbol{w}_t)\boldsymbol{h}_t - B_t\,\alpha\nabla f_t(\boldsymbol{w}_t) & -\gamma\alpha\frac{\mathrm{d}\,(\nabla^2 f_t(\boldsymbol{w}_t)\boldsymbol{h}_t)}{d\boldsymbol{w}_t} - B_t\,\alpha\nabla^2 f_t(\boldsymbol{w}_t) & \gamma\big(I - \alpha\nabla^2 f_t(\boldsymbol{w}_t)\big) \end{array} \right].
$$

### A.3.1  Base AdamW

The base update according to the AdamW algorithm (Loizou et al., 2021) is as follows,

$$
\boldsymbol{m}_{t+1} = \rho\,\boldsymbol{m}_t + \nabla f_t(\boldsymbol{w}_t),
$$
$$
\boldsymbol{v}_{t+1} = \lambda\,\boldsymbol{v}_t + \nabla f_t(\boldsymbol{w}_t)^2,
$$
$$
\mu_t = \left(\frac{1-\rho}{1-\rho^t}\right) \Big/ \sqrt{\frac{1-\lambda}{1-\lambda^t}}, \tag{58}
$$
$$
\boldsymbol{w}_{t+1} = \boldsymbol{w}_t - \boldsymbol{\alpha}_t \mu_t \frac{\boldsymbol{m}_t}{\sqrt{\boldsymbol{v}_t}} - \kappa \boldsymbol{\alpha}_t \boldsymbol{w}_t,
$$

where $\boldsymbol{m}_t$ is the momentum vector, $\boldsymbol{v}_t$ is the trace of gradient square used for normalization, and $\kappa > 0$ is a weight-decay parameter. Therefore the base algorithm needs to keep track of $\boldsymbol{w}_t, \boldsymbol{m}_t, \boldsymbol{v}_t$, i.e.,

$$
\boldsymbol{x}_t = \left[ \begin{array}{c} \boldsymbol{w}_t \\ \boldsymbol{m}_t \\ \boldsymbol{v}_t \end{array} \right]. \tag{59}
$$

It then follows from (45) that

$$
G_t^{\text{base}} = \left[ \begin{array}{cccc} \frac{\mathrm{d}\,\boldsymbol{x}_{t+1}}{\mathrm{d}\,\boldsymbol{\beta}_t} & 0 & \frac{\mathrm{d}\,\boldsymbol{x}_{t+1}}{\mathrm{d}\,\boldsymbol{x}_t} & 0 \\ \frac{\mathrm{d}\,\boldsymbol{h}_{t+1}}{\mathrm{d}\,\boldsymbol{\beta}_t} & 0 & \frac{\mathrm{d}\,\boldsymbol{h}_{t+1}}{\mathrm{d}\,\boldsymbol{x}_t} & \frac{\mathrm{d}\,\boldsymbol{h}_{t+1}}{\mathrm{d}\,\boldsymbol{h}_t} \end{array} \right]
$$

$$
= \left[ \begin{array}{cc|ccc|c} \frac{\mathrm{d}\,\boldsymbol{w}_{t+1}}{\mathrm{d}\,\boldsymbol{\beta}_t} & 0 & \frac{\mathrm{d}\,\boldsymbol{w}_{t+1}}{\mathrm{d}\,\boldsymbol{w}_t} & \frac{\mathrm{d}\,\boldsymbol{w}_{t+1}}{\mathrm{d}\,\boldsymbol{m}_t} & \frac{\mathrm{d}\,\boldsymbol{w}_{t+1}}{\mathrm{d}\,\boldsymbol{v}_t} & 0 \\ \frac{\mathrm{d}\,\boldsymbol{m}_{t+1}}{\mathrm{d}\,\boldsymbol{\beta}_t} & 0 & \frac{\mathrm{d}\,\boldsymbol{m}_{t+1}}{\mathrm{d}\,\boldsymbol{w}_t} & \frac{\mathrm{d}\,\boldsymbol{m}_{t+1}}{\mathrm{d}\,\boldsymbol{m}_t} & \frac{\mathrm{d}\,\boldsymbol{m}_{t+1}}{\mathrm{d}\,\boldsymbol{v}_t} & 0 \\ \frac{\mathrm{d}\,\boldsymbol{v}_{t+1}}{\mathrm{d}\,\boldsymbol{\beta}_t} & 0 & \frac{\mathrm{d}\,\boldsymbol{v}_{t+1}}{\mathrm{d}\,\boldsymbol{w}_t} & \frac{\mathrm{d}\,\boldsymbol{v}_{t+1}}{\mathrm{d}\,\boldsymbol{m}_t} & \frac{\mathrm{d}\,\boldsymbol{v}_{t+1}}{\mathrm{d}\,\boldsymbol{v}_t} & 0 \\ \hline \frac{\mathrm{d}\,\boldsymbol{h}_{t+1}}{\mathrm{d}\,\boldsymbol{\beta}_t} & 0 & \frac{\mathrm{d}\,\boldsymbol{h}_{t+1}}{\mathrm{d}\,\boldsymbol{w}_t} & \frac{\mathrm{d}\,\boldsymbol{h}_{t+1}}{\mathrm{d}\,\boldsymbol{m}_t} & \frac{\mathrm{d}\,\boldsymbol{h}_{t+1}}{\mathrm{d}\,\boldsymbol{v}_t} & \frac{\mathrm{d}\,\boldsymbol{h}_{t+1}}{\mathrm{d}\,\boldsymbol{h}_t} \end{array} \right]
$$

$$
= \left[ \begin{array}{cc|ccc|c} -\mu_t\left[\frac{\boldsymbol{m}_t}{\sqrt{\boldsymbol{v}_t}} + \kappa\boldsymbol{w}_t\right]\sigma'(\boldsymbol{\beta}_t) & 0 & I - \kappa[\boldsymbol{\alpha}_t] & -\mu_t\left[\frac{\boldsymbol{\alpha}_t}{\sqrt{\boldsymbol{v}_t}}\right] & \frac{\mu_t}{2}\left[\frac{\boldsymbol{\alpha}_t\boldsymbol{m}_t}{\boldsymbol{v}_t^{1.5}}\right] & 0 \\ 0 & 0 & \nabla^2 f_t & \rho I & 0 & 0 \\ 0 & 0 & 2[\nabla f_t]\,\nabla^2 f_t & 0 & \lambda I & 0 \\ \hline \frac{\mathrm{d}\,\boldsymbol{h}_{t+1}}{\mathrm{d}\,\boldsymbol{\beta}_t} & 0 & \frac{\mathrm{d}\,\boldsymbol{h}_{t+1}}{\mathrm{d}\,\boldsymbol{w}_t} & \frac{\mathrm{d}\,\boldsymbol{h}_{t+1}}{\mathrm{d}\,\boldsymbol{m}_t} & \frac{\mathrm{d}\,\boldsymbol{h}_{t+1}}{\mathrm{d}\,\boldsymbol{v}_t} & \frac{\mathrm{d}\,\boldsymbol{h}_{t+1}}{\mathrm{d}\,\boldsymbol{h}_t} \end{array} \right] \tag{60}
$$

where the last equality follows from simple derivative computations in (58).

We proceed to compute the terms in the last row of the $G_t^{\text{base}}$ above. Consider the following block representation of $X_t$,

$$
X_t = \left[ \begin{array}{c} \mathcal{H}_t \\ X_t^m \\ X_t^v \end{array} \right], \tag{61}
$$

Plugging the first row of $G_t^{\text{base}}$, given in (60), into (47), implies that

$$\mathcal{H}_{t+1} = -\gamma\mu_t\left[\frac{\boldsymbol{m}_t}{\sqrt{\boldsymbol{v}_t}} + \kappa\boldsymbol{w}_t\right]\sigma'(\boldsymbol{\beta}_t)B_t + \gamma\big(I - \kappa\left[\boldsymbol{\alpha}_t\right]\big)\mathcal{H}_t - \gamma\mu_t\left[\frac{\boldsymbol{\alpha}_t}{\sqrt{\boldsymbol{v}_t}}\right]X_t^m + \gamma\frac{\mu_t}{2}\left[\frac{\boldsymbol{\alpha}_t\boldsymbol{m}_t}{\boldsymbol{v}_t^{1.5}}\right]X_t^v.$$

(62)

for all $t \geq 0$. Note that for any pair of same-size vectors $\boldsymbol{a}$ and $\boldsymbol{b}$, we have $[\boldsymbol{a}]\,\boldsymbol{b} = [\boldsymbol{b}]\,\boldsymbol{a}$ where $[\boldsymbol{a}]$ and $[\boldsymbol{b}]$ are diagonal matrices of $\boldsymbol{a}$ and $\boldsymbol{b}$, respectively. Therefore, the $i$th column in the matrix equation (62) can be equivalently written as

$$\mathcal{H}_{t+1}^{[i]} = -\gamma\mu_t\left[\sigma'(\boldsymbol{\beta}_t)B_t^{[i]}\right]\frac{\boldsymbol{m}_t}{\sqrt{\boldsymbol{v}_t}} + \kappa\boldsymbol{w}_t + \gamma\big(I - \kappa\left[\boldsymbol{\alpha}_t\right]\big)\mathcal{H}_t^{[i]} - \gamma\mu_t\left[X_t^{m\,[i]}\right]\frac{\boldsymbol{\alpha}_t}{\sqrt{\boldsymbol{v}_t}} + \gamma\frac{\mu_t}{2}\left[X_t^{v\,[i]}\right]\frac{\boldsymbol{\alpha}_t\boldsymbol{m}_t}{\boldsymbol{v}_t^{1.5}},$$

(63)

where $B_t^{[i]}$, $\mathcal{H}_t^{[i]}$, $X_t^{m\,[i]}$, and $X_t^{v\,[i]}$ stand for the $i$th columns of $B_t$, $\mathcal{H}_t$, $X_t^m$, and $X_t^v$, respectively. Following similar arguments as in (48), it is easy to show that

$$\frac{\mathrm{d}\,X_t^m}{\mathrm{d}\,\boldsymbol{\beta}_t} = \frac{\mathrm{d}\,X_t^v}{\mathrm{d}\,\boldsymbol{\beta}_t} = 0,$$
$$\frac{\mathrm{d}\,X_t^m}{\mathrm{d}\,\boldsymbol{w}_t} = \frac{\mathrm{d}\,X_t^v}{\mathrm{d}\,\boldsymbol{w}_t} = 0,$$
$$\frac{\mathrm{d}\,X_t^m}{\mathrm{d}\,\boldsymbol{m}_t} = \frac{\mathrm{d}\,X_t^v}{\mathrm{d}\,\boldsymbol{m}_t} = 0,$$
$$\frac{\mathrm{d}\,X_t^m}{\mathrm{d}\,\boldsymbol{v}_t} = \frac{\mathrm{d}\,X_t^v}{\mathrm{d}\,\boldsymbol{v}_t} = 0,$$
$$\frac{\mathrm{d}\,X_t^m}{\mathrm{d}\,\boldsymbol{h}_t} = \frac{\mathrm{d}\,X_t^v}{\mathrm{d}\,\boldsymbol{h}_t} = 0.$$

(64)

Note that $\boldsymbol{h}_t$ is an $nm$-dimensional vector derived from stacking the columns of $\mathcal{H}_t$. Therefore, we consider a block representation of $\boldsymbol{h}_t$ consisting of $m$ blocks, each of which corresponds to a column of $\mathcal{H}_t$. By taking the derivative of (62) with respect to $\boldsymbol{h}_t$, and using (64), we obtain

$$\frac{\mathrm{d}\,\boldsymbol{h}_{t+1}}{\mathrm{d}\,\boldsymbol{h}_t} = \gamma\begin{bmatrix} I - \kappa\left[\boldsymbol{\alpha}_t\right] & 0 & 0 & 0 \\ \hline 0 & I - \kappa\left[\boldsymbol{\alpha}_t\right] & 0 & 0 \\ \hline 0 & 0 & \ddots & 0 \\ \hline 0 & 0 & 0 & I - \kappa\left[\boldsymbol{\alpha}_t\right] \end{bmatrix}\begin{matrix} \leftarrow 1\text{st} \\ \\ \leftarrow 2\text{nd} \\ \\ \vdots \\ \\ \leftarrow m\text{th} \end{matrix}.$$

(65)

Let $\beta_t[i]$ and $w_t[j]$ denote the $i$th and $j$th entries of $\boldsymbol{\beta}_t$ and $\boldsymbol{w}_t$, for $i = 1, \ldots, m$ and $j = 1, \ldots, n$, respectively. Note that $\mathrm{d}\,\boldsymbol{h}_{t+1}/\mathrm{d}\,\boldsymbol{\beta}_t$ is a block matrix, in the form of an $m \times m$ array of $n \times 1$ blocks, $\frac{\mathrm{d}\,\boldsymbol{h}_{t+1}}{\mathrm{d}\,\boldsymbol{\beta}_t}[i,j] \overset{\text{def}}{=} \frac{\mathrm{d}\,\mathcal{H}_{t+1}^{[i]}}{\mathrm{d}\,\beta_t[j]}$, for $i, j = 1, \ldots, m$. It then follows from (62) and (64) that, for $i, j = 1, \ldots, m$,

$$\frac{\mathrm{d}\,\boldsymbol{h}_{t+1}}{\mathrm{d}\,\boldsymbol{\beta}_t}[i,j] = \frac{\mathrm{d}\,\mathcal{H}_{t+1}^{[i]}}{\mathrm{d}\,\beta_t[j]}$$

$$= -\gamma\mu_t\left[\frac{\boldsymbol{m}_t}{\sqrt{\boldsymbol{v}_t}} + \kappa\boldsymbol{w}_t\right]\left(\frac{\partial\,\sigma'(\boldsymbol{\beta}_t)}{\partial\,\beta_t[j]}\right)B_t^{[i]} + \gamma\left(I - \kappa\left[\frac{\mathrm{d}\,\boldsymbol{\alpha}_t}{\mathrm{d}\,\beta_t[j]}\right]\right)\mathcal{H}_t^{[i]}$$

$$- \gamma\mu_t\left[\frac{1}{\sqrt{\boldsymbol{v}_t}}\right]\left[\frac{\mathrm{d}\,\boldsymbol{\alpha}_t}{\mathrm{d}\,\beta_t[j]}\right]X_t^{m\,[i]} + \gamma\frac{\mu_t}{2}\left[\frac{\boldsymbol{m}_t}{\boldsymbol{v}_t^{1.5}}\right]\left[\frac{\mathrm{d}\,\boldsymbol{\alpha}_t}{\mathrm{d}\,\beta_t[j]}\right]X_t^{v\,[i]},$$

(66)

where $\frac{\partial}{\partial\beta}$ stands for the entry-wise partial derivative of a matrix with respect to a scalar variable $\beta$.

In the same vein, it follows from (63) and (64) that

$$\frac{\mathrm{d}\,\boldsymbol{h}_{t+1}}{\mathrm{d}\,\boldsymbol{w}_t} = -\gamma\mu_t\kappa\begin{bmatrix} \left[\sigma'(\boldsymbol{\beta}_t)B_t^{[1]}\right] \\ \hline \vdots \\ \hline \left[\sigma'(\boldsymbol{\beta}_t)B_t^{[m]}\right] \end{bmatrix},$$

(67)

$$\frac{\mathrm{d}\,\boldsymbol{h}_{t+1}}{\mathrm{d}\,\boldsymbol{m}_t} = \gamma\mu_t \begin{bmatrix} \left[ \frac{\boldsymbol{\alpha}_t X_t^{v\,[1]}}{2\,\boldsymbol{v}_t^{1.5}} - \frac{\sigma'(\boldsymbol{\beta}_t)B_t^{[1]}}{\sqrt{\boldsymbol{v}_t}} \right] \\ \vdots \\ \left[ \frac{\boldsymbol{\alpha}_t X_t^{v\,[m]}}{2\,\boldsymbol{v}_t^{1.5}} - \frac{\sigma'(\boldsymbol{\beta}_t)B_t^{[m]}}{\sqrt{\boldsymbol{v}_t}} \right] \end{bmatrix}, \tag{68}$$

$$\frac{\mathrm{d}\,\boldsymbol{h}_{t+1}}{\mathrm{d}\,\boldsymbol{v}_t} = \frac{\gamma\mu_t}{2} \begin{bmatrix} \left[ \frac{1}{\boldsymbol{v}_t^{1.5}} \right] \left[ \left(\sigma'(\boldsymbol{\beta}_t)B_t^{[1]}\right)\boldsymbol{m}_t + \boldsymbol{\alpha}_t X_t^{m\,[1]} - \frac{3\boldsymbol{\alpha}_t \boldsymbol{m}_t X_t^{v\,[1]}}{2\,\boldsymbol{v}_t} \right] \\ \vdots \\ \left[ \frac{1}{\boldsymbol{v}_t^{1.5}} \right] \left[ \left(\sigma'(\boldsymbol{\beta}_t)B_t^{[m]}\right)\boldsymbol{m}_t + \boldsymbol{\alpha}_t X_t^{m\,[m]} - \frac{3\boldsymbol{\alpha}_t \boldsymbol{m}_t X_t^{v\,[m]}}{2\,\boldsymbol{v}_t} \right] \end{bmatrix}. \tag{69}$$

Finally, $G_t^{\text{base}}$ is obtained by plugging (65), (66), (67), (68), and (69) into (60).

### A.3.2 Base Lion

The lion algorithm, when used for base update, is as follows

$$\boldsymbol{m}_{t+1} = \rho\,\boldsymbol{m}_t + (1-\rho)\,\nabla f_t(\boldsymbol{w}_t), \tag{70}$$

$$\boldsymbol{w}_{t+1} = \boldsymbol{w}_t - \boldsymbol{\alpha}_t\,\mathrm{Sign}\left(c\,\boldsymbol{m}_t + (1-c)\nabla f_t\right) - \kappa\boldsymbol{\alpha}_t\boldsymbol{w}_t, \tag{71}$$

where $\boldsymbol{m}_t$ is called the momentum, $\kappa > 0$ is the weight-decay parameter, $\rho, c \in [0, 1)$ are constants, and $\mathrm{Sign}(\cdot)$ is a function that computes entry-wise sign of a vector. Let

$$\boldsymbol{x}_t = \begin{bmatrix} \boldsymbol{w}_t \\ \boldsymbol{m}_t \end{bmatrix}. \tag{72}$$

It then follows from (45) that

$$\begin{aligned}
G_t^{\text{base}} &= \begin{bmatrix} \frac{\mathrm{d}\,\boldsymbol{x}_{t+1}}{\mathrm{d}\,\boldsymbol{\beta}_t} & 0 & \frac{\mathrm{d}\,\boldsymbol{x}_{t+1}}{\mathrm{d}\,\boldsymbol{x}_t} & 0 \\ \frac{\mathrm{d}\,\boldsymbol{h}_{t+1}}{\mathrm{d}\,\boldsymbol{\beta}_t} & 0 & \frac{\mathrm{d}\,\boldsymbol{h}_{t+1}}{\mathrm{d}\,\boldsymbol{x}_t} & \frac{\mathrm{d}\,\boldsymbol{h}_{t+1}}{\mathrm{d}\,\boldsymbol{h}_t} \end{bmatrix} \\
&= \begin{bmatrix} \frac{\mathrm{d}\,\boldsymbol{w}_{t+1}}{\mathrm{d}\,\boldsymbol{\beta}_t} & 0 & \frac{\mathrm{d}\,\boldsymbol{w}_{t+1}}{\mathrm{d}\,\boldsymbol{w}_t} & \frac{\mathrm{d}\,\boldsymbol{w}_{t+1}}{\mathrm{d}\,\boldsymbol{m}_t} & 0 \\ \frac{\mathrm{d}\,\boldsymbol{m}_{t+1}}{\mathrm{d}\,\boldsymbol{\beta}_t} & 0 & \frac{\mathrm{d}\,\boldsymbol{m}_{t+1}}{\mathrm{d}\,\boldsymbol{w}_t} & \frac{\mathrm{d}\,\boldsymbol{m}_{t+1}}{\mathrm{d}\,\boldsymbol{m}_t} & 0 \\ \frac{\mathrm{d}\,\boldsymbol{h}_{t+1}}{\mathrm{d}\,\boldsymbol{\beta}_t} & 0 & \frac{\mathrm{d}\,\boldsymbol{h}_{t+1}}{\mathrm{d}\,\boldsymbol{w}_t} & \frac{\mathrm{d}\,\boldsymbol{h}_{t+1}}{\mathrm{d}\,\boldsymbol{m}_t} & \frac{\mathrm{d}\,\boldsymbol{h}_{t+1}}{\mathrm{d}\,\boldsymbol{h}_t} \end{bmatrix} \\
&= \begin{bmatrix} -\left[\mathrm{Sign}\left(c\,\boldsymbol{m}_t + (1-c)\nabla f_t\right) + \kappa\boldsymbol{w}_t\right]\sigma'(\boldsymbol{\beta}_t) & 0 & I - \kappa\,[\boldsymbol{\alpha}_t] & 0 & 0 \\ \frac{\mathrm{d}\,\boldsymbol{m}_{t+1}}{\mathrm{d}\,\boldsymbol{\beta}_t} & 0 & \frac{\mathrm{d}\,\boldsymbol{m}_{t+1}}{\mathrm{d}\,\boldsymbol{w}_t} & \frac{\mathrm{d}\,\boldsymbol{m}_{t+1}}{\mathrm{d}\,\boldsymbol{m}_t} & 0 \\ \frac{\mathrm{d}\,\boldsymbol{h}_{t+1}}{\mathrm{d}\,\boldsymbol{\beta}_t} & 0 & \frac{\mathrm{d}\,\boldsymbol{h}_{t+1}}{\mathrm{d}\,\boldsymbol{w}_t} & \frac{\mathrm{d}\,\boldsymbol{h}_{t+1}}{\mathrm{d}\,\boldsymbol{m}_t} & \frac{\mathrm{d}\,\boldsymbol{h}_{t+1}}{\mathrm{d}\,\boldsymbol{h}_t} \end{bmatrix}
\end{aligned} \tag{73}$$

where the second equality is due to (72) and the last equality follows from (71). Consider the following block representation of $X_t$,

$$X_t = \begin{bmatrix} \mathcal{H}_t \\ X_t^m \end{bmatrix}. \tag{74}$$

Plugging the first row of $G_t^{\text{base}}$, given in (73), into (47), implies that

$$\mathcal{H}_{t+1} = -\gamma\left[\mathrm{Sign}\left(c\,\boldsymbol{m}_t + (1-c)\nabla f_t\right) + \kappa\boldsymbol{w}_t\right]\sigma'(\boldsymbol{\beta}_t)\,B_t + \gamma\left(I - \kappa\,[\boldsymbol{\alpha}_t]\right)\mathcal{H}_t \tag{75}$$

For simplicity of notation, we define the diagonal matrix $S_t$ as

$$S_t \stackrel{\text{def}}{=} \left[\mathrm{Sign}\left(c\,\boldsymbol{m}_t + (1-c)\nabla f_t\right) + \kappa\boldsymbol{w}_t\right]. \tag{76}$$

Then,

$$\boldsymbol{h}_{t+1} = \gamma \begin{bmatrix} -S_t\,\sigma'(\boldsymbol{\beta}_t)\,B_t^{[1]} + \gamma\left(I - \kappa\,[\boldsymbol{\alpha}_t]\right)\mathcal{H}_t^{[1]} \\ \vdots \\ -S_t\,\sigma'(\boldsymbol{\beta}_t)\,B_t^{[m]} + \gamma\left(I - \kappa\,[\boldsymbol{\alpha}_t]\right)\mathcal{H}_t^{[m]} \end{bmatrix} \tag{77}$$

It follows that

$$\frac{\mathrm{d}\, \boldsymbol{h}_{t+1}}{\mathrm{d}\, \boldsymbol{m}_t} = 0, \tag{78}$$

and

$$\frac{\mathrm{d}\, \boldsymbol{h}_{t+1}}{\mathrm{d}\, \boldsymbol{w}_t} = -\gamma \left[ \begin{array}{c|c|c} [\boldsymbol{e}_1]\; \sigma'(\boldsymbol{\beta}_t)\, B_t^{[1]} & \cdots & [\boldsymbol{e}_n]\; \sigma'(\boldsymbol{\beta}_t)\, B_t^{[1]} \\ \hline \vdots & \ddots & \vdots \\ \hline [\boldsymbol{e}_1]\; \sigma'(\boldsymbol{\beta}_t)\, B_t^{[m]} & \cdots & [\boldsymbol{e}_n]\; \sigma'(\boldsymbol{\beta}_t)\, B_t^{[m]} \end{array} \right], \tag{79}$$

where $\boldsymbol{e}_i$ is the $i$th unit vector (i.e., an $n$-dimensional vector whose $i$th entry is 1 and all other entries are zero). Let $\beta_t[i]$ and $\mathcal{H}_t^{[i]}$ be the $i$th entry of $\boldsymbol{\beta}_t$ and $i$th column of $\mathcal{H}_t$, respectively, for $i = 1, \ldots, m$. Then,

$$\frac{\mathrm{d}\, \boldsymbol{h}_{t+1}}{\mathrm{d}\, \boldsymbol{\beta}_t} = -\gamma \left[ \begin{array}{c|c|c} \gamma\kappa \left[\frac{\mathrm{d}\, \boldsymbol{\alpha}_t}{\mathrm{d}\, \beta_t[1]}\right] \mathcal{H}_t^{[1]} + S_t \frac{\partial\, \sigma'(\boldsymbol{\beta}_t)}{\partial\, \beta_t[1]} B_t^{[1]} & \cdots & \gamma\kappa \left[\frac{\mathrm{d}\, \boldsymbol{\alpha}_t}{\mathrm{d}\, \beta_t[m]}\right] \mathcal{H}_t^{[1]} + S_t \frac{\partial\, \sigma'(\boldsymbol{\beta}_t)}{\partial\, \beta_t[m]} B_t^{[1]} \\ \hline \vdots & \ddots & \vdots \\ \hline \gamma\kappa \left[\frac{\mathrm{d}\, \boldsymbol{\alpha}_t}{\mathrm{d}\, \beta_t[1]}\right] \mathcal{H}_t^{[m]} + S_t \frac{\partial\, \sigma'(\boldsymbol{\beta}_t)}{\partial\, \beta_t[1]} B_t^{[m]} & \cdots & \gamma\kappa \left[\frac{\mathrm{d}\, \boldsymbol{\alpha}_t}{\mathrm{d}\, \beta_t[m]}\right] \mathcal{H}_t^{[m]} + S_t \frac{\partial\, \sigma'(\boldsymbol{\beta}_t)}{\partial\, \beta_t[m]} B_t^{[m]} \end{array} \right], \tag{80}$$

and

$$\frac{\mathrm{d}\, \boldsymbol{h}_{t+1}}{\mathrm{d}\, \boldsymbol{h}_t} = \gamma \left[ \begin{array}{c|c|c|c} I - \kappa\,[\boldsymbol{\alpha}_t] & 0 & 0 & 0 \\ \hline 0 & I - \kappa\,[\boldsymbol{\alpha}_t] & 0 & 0 \\ \hline 0 & 0 & \ddots & 0 \\ \hline 0 & 0 & 0 & I - \kappa\,[\boldsymbol{\alpha}_t] \end{array} \right] \begin{array}{l} \leftarrow 1\text{st} \\[6pt] \leftarrow 2\text{nd} \\[6pt] \vdots \\[6pt] \leftarrow m\text{th} \end{array}. \tag{81}$$

It follows from (22), (73), and (78) that in the $G_t$ matrix, $\frac{\mathrm{d}\, \boldsymbol{m}_{t+1}}{\mathrm{d}\, \boldsymbol{m}_t}$ is the only non-zero block in its corresponding column of blocks. Consequently, it follows from (15) that $X_t^m$, as defined in (74), has no impact on the update of $\mathcal{H}_{t+1}$, $Y_{t+1}$, and $Q_{t+1}$. Therefore, the rows and columns of $G^{\text{base}}$ that correspond to derivative of $\boldsymbol{m}$ can be completely removed from $G^{\text{base}}$. By removing these rows and columns from $G^t$, the matrix update (15) simplifies to

$$\left[ \begin{array}{c} Y_{t+1} \\ \mathcal{H}_{t+1} \\ Q_{t+1} \end{array} \right] = \gamma \left[ \begin{array}{cc} \left[ -\big[\operatorname{Sign}\big(c\,\boldsymbol{m}_t + (1-c)\nabla f_t\big)\big]\sigma'(\boldsymbol{\beta}_t) \quad 0 \right] & \begin{array}{cc} \frac{\mathrm{d}\, \boldsymbol{y}_{t+1}}{\mathrm{d}\, \boldsymbol{y}_t} & \frac{\mathrm{d}\, \boldsymbol{y}_{t+1}}{\mathrm{d}\, \boldsymbol{w}_t} \quad \frac{\mathrm{d}\, \boldsymbol{y}_{t+1}}{\mathrm{d}\, \boldsymbol{h}_t} \\ I - \kappa\,[\boldsymbol{\alpha}_t] \quad 0 \end{array} \\ \left[ \frac{\mathrm{d}\, \boldsymbol{h}_{t+1}}{\mathrm{d}\, \boldsymbol{\beta}_t} \quad 0 \right] & \begin{array}{cc} \frac{\mathrm{d}\, \boldsymbol{h}_{t+1}}{\mathrm{d}\, \boldsymbol{w}_t} & \frac{\mathrm{d}\, \boldsymbol{h}_{t+1}}{\mathrm{d}\, \boldsymbol{h}_t} \end{array} \end{array} \right] \left( \left[ \begin{array}{c} Y_t \\ \mathcal{H}_t \\ Q_t \end{array} \right] + (1-\gamma) \left[ \begin{array}{c} I \\ 0 \\ 0 \\ 0 \end{array} \right] \right), \tag{82}$$

where $\mathrm{d}\, \boldsymbol{h}_{t+1}/\mathrm{d}\, \boldsymbol{\beta}_t$, $\mathrm{d}\, \boldsymbol{h}_{t+1}/\mathrm{d}\, \boldsymbol{w}_t$, and $\mathrm{d}\, \boldsymbol{h}_{t+1}/\mathrm{d}\, \boldsymbol{h}_t$ are given in (80), (79), and (81), respectively; and the blocks in the first row depend on the meta update.

## B  EXITING STEP-SIZE OPTIMIZATION ALGORITHMS AS SPECIAL CASES OF METAOPTIMIZE

In this appendix we show that some of the existing step-size optimization algorithms are special cases of the MetaOptimize framework. In particular, we first consider the IDBD algorithm (Sutton, 1992) and its extension (Xu et al., 2018), and then discuss about the HyperGradient algorithm (Baydin et al., 2017).

### B.1  IDBD AND ITS EXTENSIONS

Sutton (1992) proposed the IDBD algorithm for step-size optimization of a class of quadratic loss functions. In particular, it considers loss functions of the form

$$f_t(\boldsymbol{w}_t) = \frac{1}{2}\big(\boldsymbol{a}_t^T \boldsymbol{w}_t - b_t\big)^2, \tag{83}$$

for a given sequence of feature vectors $\boldsymbol{a}_t$ and target values $b_t$, for $t = 1, 2, \ldots$. Moreover, Sutton (1992) assumes weight-wise step sizes, in which case $\boldsymbol{\beta}_t$ has the same dimension as $\boldsymbol{w}_t$. The update rule of IDBD is as follows:

$$\boldsymbol{g}_t \leftarrow \left(\boldsymbol{a}_t^T \boldsymbol{w}_t - b_t\right)\boldsymbol{a}_t, \tag{84}$$

$$\boldsymbol{\beta}_{t+1} \leftarrow \boldsymbol{\beta}_t - \eta\,\boldsymbol{h}_t\,\boldsymbol{g}_t, \tag{85}$$

$$\boldsymbol{\alpha}_{t+1} \leftarrow \exp\left(\boldsymbol{\beta}_{t+1}\right), \tag{86}$$

$$\boldsymbol{w}_{t+1} \leftarrow \boldsymbol{w}_t - \boldsymbol{\alpha}_{t+1}\,\boldsymbol{g}_t, \tag{87}$$

$$\boldsymbol{h}_{t+1} \leftarrow \left(1 - \boldsymbol{\alpha}_{t+1}\boldsymbol{a}_t^2\right)^+ \boldsymbol{h}_t - \boldsymbol{\alpha}_{t+1}\boldsymbol{g}_t, \tag{88}$$

where $(\cdot)^+$ clips the entries at zero to make them non-negative, aimed to improve stability. Here, $\boldsymbol{g}_t$ is the gradient of $f_t(\boldsymbol{w}_t)$ and $\boldsymbol{a}_t^2$ in the last line is a vector that contains diagonal entries of the Hessian of $f_t$. The updated values of $\boldsymbol{\beta}$ and $\boldsymbol{w}$ would remain unchanged, if instead of the vector $\boldsymbol{h}_t$, we use a diagonal matrix $\mathcal{H}_t$ and replace (85) and (88) by

$$\boldsymbol{\beta}_{t+1} \leftarrow \boldsymbol{\beta}_t - \eta\,\mathcal{H}_t\,\boldsymbol{g}_t,$$
$$\mathcal{H}_{t+1} \leftarrow \left(1 - \left[\boldsymbol{\alpha}_{t+1}\boldsymbol{a}^2\right]\right)^+ \mathcal{H}_t - \left[\boldsymbol{\alpha}_{t+1}\boldsymbol{g}_t\right]. \tag{89}$$

Note that $\left[\boldsymbol{a}^2\right]$ is a matrix that is obtained from zeroing-out all non-diagonal entries of the Hessian matrix of $f_t$. It is easy to see that the above formulation of IDBD, equals the L-approximation of MetaOptimize framework when we use SGD for both base and meta updates, and further use a diagonal approximation of the Hessian matrix along with a rectifier in the update of $\mathcal{H}_t$.

An extension of IDBD beyond quadratic case has been derived in (Xu et al., 2018). Similar to IDBD, they also consider weight-wise step sizes, i.e., $m = n$. The update of step sizes in this method is as follows:

$$\boldsymbol{\beta}_{t+1} \leftarrow \boldsymbol{\beta}_t - \eta\,\mathcal{H}_t^\mathsf{T}\,\nabla f_t(\boldsymbol{w}_t)$$
$$\boldsymbol{\alpha}_{t+1} \leftarrow \exp(\boldsymbol{\beta}_{t+1}),$$
$$\boldsymbol{w}_{t+1} \leftarrow \boldsymbol{w}_t - \boldsymbol{\alpha}_{t+1}\,\nabla f_t(\boldsymbol{w}_t),$$
$$\mathcal{H}_{t+1} \leftarrow \left(I - \left[\boldsymbol{\alpha}_{t+1}\right]\nabla^2 f_t(\boldsymbol{w}_t)\right)\mathcal{H}_t - \left[\boldsymbol{\alpha}_{t+1}\nabla f_t(\boldsymbol{w}_t)\right].$$

Similar to IDBD, it is straightforward to check that the above set of updates is equivalent to the L-approximation of MetaOptimize framework that uses SGD for both base and meta updates, except for the fact that the above algorithm uses $\boldsymbol{\alpha}_{t+1}$ in $\boldsymbol{w}_{t+1}$ and $\mathcal{H}_{t+1}$ updates whereas MetaOptimize uses $\boldsymbol{\alpha}_t$. This however has no considerable impact since $\boldsymbol{\alpha}_t$ varies slowly.

### B.2 HYPER-GRADIENT DESCENT

HyperGradient descent was proposed in (Baydin et al., 2017) as a step-size optimization method. It considers scalar step size with straightforward extensions to weight-wise step sizes, and at each time $t$, updates the step size in a direction to minimize the immediate next loss function. In particular, they propose the following additive update for step sizes, that can wrap around an arbitrary base update:

$$\boldsymbol{\alpha}_t = \beta_t\,\mathbf{1}_{n \times 1},$$
$$\beta_{t+1} = \beta_t - \eta\,\frac{\mathrm{d}\,f_t(\boldsymbol{w}_t)}{\mathrm{d}\,\beta_{t-1}} = \beta_t - \eta\,\nabla f_t(\boldsymbol{w}_t)^T\,\frac{\mathrm{d}\,\boldsymbol{w}_t}{\mathrm{d}\,\beta_{t-1}}. \tag{90}$$

The last update can be equivalently written as

$$\beta_{t+1} = \beta_t - \eta\,\mathcal{H}_t^T\,\nabla f_t(\boldsymbol{w}_t),$$
$$\mathcal{H}_{t+1} = 0 \times \mathcal{H}_t + \frac{\mathrm{d}\,\boldsymbol{w}_{t+1}}{\mathrm{d}\,\beta_t}. \tag{91}$$

The step-size update in (91) can be perceived as a special case of MetaOptimize in two different ways. First, as a MetaOptimize algorithm that uses SGD as its meta update and approximate the $G_t$ matrix in (10) by zeroing out all of its blocks except for the top two blocks in the first column. From another perspective, the additive HyperGradient descent in (91) is also equivalent to a MetaOptimize algorithm that uses SGD as its meta update and sets $\gamma = 0$. Note that setting $\gamma$ equal to zero would eliminate the dependence of $\mathcal{H}_{t+1}$ on $X_t$ and $Q_t$, as can be verified from (15). This would also render the $\beta$ updates ignorant about the long-term impact of step size on future losses.

## C    EXPERIMENT DETAILS

In the appendix, we describe the details of experiments performed throughout the paper. In our experiments on CIFAR10 and ImageNet dataset, we used a machine with four Intel Xeon Gold 5120 Skylake @ 2.2GHz CPUs and a single NVIDIA V100 Volta (16GB HBM2 memory) GPU. For TinyStories dataset, we used a machine with four AMD Milan 7413 @ 2.65 GHz 128M cache L3 CPUs and a single NVIDIA A100SXM4 (40 GB memory) GPU. In all experiments, the meta step size $\eta$ is set to $10^{-3}$. The meta-parameters used in the considered optimization algorithm for CIFAR10, ImageNet, and TinyStories are given in Table 2, Table 3, and Table 4, respectively. In the experiments, we performed a grid search for $\rho, \bar{\rho} \in \{0.9, 0.99, 0.999\}$, $\lambda, \bar{\lambda} \in \{0.99, 0.999\}$, and $c, \bar{c} \in \{0.9, 0.99\}$. Regarding baselines with fixed step sizes, we did a grid search for the learning rate in the set $\{10^{-5}, 10^{-4}, 10^{-3}, 10^{-2}, 10^{-1}\}$. We set $\gamma$ equal to one in all experiments. Moreover, in ImageNet (respectively TinyStories) dataset, for AdamW with the learning rate scheduler, we considered a cosine decay with 10k (respectively 1k) steps warmup (according to extensive experimental studies in (Chen et al., 2023) (respectively (Karpathy, 2024))) and did a grid search for the maximum learning rate in the set $\{10^{-5}, 10^{-4}, 10^{-3}\}$.

Regarding other baseline algorithms, for DoG, although it is a parameter-free algorithm, its performance is still sensitive to the initial step movement. We did a grid search for the initial step movement in the set $\{10^{-9}, 10^{-8}, 10^{-7}, 10^{-6}\}$ and reported the performance for the best value. In all experiments of DoG, we considered the polynomial decay averaging. For Prodigy, we used the default values of parameters as suggested by the authors in github repository. For gdtuo, we considered the following (base, meta) combinations: (RMSprop, Adam), (Adam, Adam), and (SGD with momentum, Adam) and chose the best combination. For mechanic, we did experiments for the base updates of SGDm, Lion, and Adam and considered the best update. In order to have a fair comparison, in mechanic and gdtuo, we used the same initial step size as MetaOptimize.

Regarding the complexity overheads reported in Table 1, for AdamW with fixed step-size we used the Pytroch implementation of AdamW. For all other baselines, we used the implementation from the Github repository provided along with (and cited in) the corresponding paper. For MetaOptimize, we used the implementation in (Anonymous, 2024). Note that the implementation of MetaOptimize in (Anonymous, 2024) is not optimized for time or space efficiency, and smaller complexity overheads might be achieved with more efficient codes. For each algorithm, the wall-clock time overhead and GPU space overhead are computed by $(T_{\text{Alg}} - T_{\text{AdamW}})/T_{\text{AdamW}}$ and $(B^{\max}_{\text{AdamW}}/B^{\max}_{\text{Alg}}) - 1$, respectively; where $T_{\text{Alg}}$ and $T_{\text{AdamW}}$ are per-iteration runtimes of the algorithm and AdamW, and $B^{\max}_{\text{Alg}}$ and $B^{\max}_{\text{AdamW}}$ are the maximum batch-sizes that did not cause GPU-memory outage for the algorithm and AdamW.

| Base Update | Meta Update (if any) | $\rho$ | $\lambda$ | $\kappa$ | $c$ | $\bar{\rho}$ | $\bar{\lambda}$ | $\bar{c}$ | $\alpha_0$ | $\eta$ | $\gamma$ |
|---|---|---|---|---|---|---|---|---|---|---|---|
| AdamW | Fixed step size | 0.9 | 0.999 | 0.1 | - | - | - | - | $10^{-5}$ | - | 1 |
| | Adam, Scalar | 0.9 | 0.999 | 0.1 | - | 0.9 | 0.999 | - | $10^{-6}$ | $10^{-3}$ | 1 |
| | Adam, Blockwise | 0.9 | 0.999 | 0.1 | - | 0.9 | 0.999 | - | $10^{-6}$ | $10^{-3}$ | 1 |
| Lion | Fixed step size | 0.99 | - | 0.1 | 0.9 | - | - | - | $10^{-4}$ | - | 1 |
| | Lion, Scalar | 0.99 | - | 0.1 | 0.9 | 0.99 | - | 0.9 | $10^{-6}$ | $10^{-3}$ | 1 |
| | Lion, Blockwise | 0.99 | - | 0.1 | 0.9 | 0.99 | - | 0.9 | $10^{-6}$ | $10^{-3}$ | 1 |
| RMSprop | Fixed step size | - | 0.999 | 0.1 | - | - | - | - | $10^{-5}$ | - | 1 |
| | Adam, Scalar | - | 0.999 | 0.1 | - | 0.9 | 0.999 | - | $10^{-6}$ | $10^{-3}$ | 1 |
| | Adam, Blockwise | - | 0.999 | 0.1 | - | 0.9 | 0.999 | - | $10^{-6}$ | $10^{-3}$ | 1 |
| SGDm | Fixed step size | 0.9 | - | 0.1 | - | - | - | - | $10^{-3}$ | - | 1 |
| | Adam, Scalar | 0.9 | - | 0.1 | - | - | - | - | $10^{-6}$ | $10^{-3}$ | 1 |
| | Adam, Blockwise | 0.9 | - | 0.1 | - | - | - | - | $10^{-6}$ | $10^{-3}$ | 1 |

Table 2: The values of meta-parameters used in CIFAR10 dataset.

## D    FURTHER EXPERIMENTAL RESULTS

**ImageNet dataset:** In Figure 6, we depict the train accuracy (top 1) and test accuracy (top 1) of the considered algorithms in ImageNet dataset. As can be seen, in the train accuracy (top

| Base Update | Meta Update | $\rho$ | $\lambda$ | $\kappa$ | $c$ | $\bar{\rho}$ | $\bar{\lambda}$ | $\bar{c}$ | $\alpha_0$ | $\eta$ | $\gamma$ |
|---|---|---|---|---|---|---|---|---|---|---|---|
| | Fixed step size | 0.9 | 0.999 | 0.1 | - | - | - | - | $10^{-5}$ | - | 1 |
| AdamW | Lion, Scalar | 0.9 | 0.999 | 0.1 | - | 0.99 | - | 0.9 | $10^{-6}$ | $10^{-3}$ | 1 |
| | Lion, Blockwise | 0.9 | 0.999 | 0.1 | - | 0.99 | - | 0.9 | $10^{-6}$ | $10^{-3}$ | 1 |
| | Fixed step size | 0.99 | - | 0.1 | 0.9 | - | - | - | $10^{-5}$ | - | 1 |
| Lion | Lion, Scalar | 0.99 | - | 0.1 | 0.9 | 0.99 | - | 0.9 | $10^{-6}$ | $10^{-3}$ | 1 |
| | Lion, Blockwise | 0.99 | - | 0.1 | 0.9 | 0.99 | - | 0.9 | $10^{-6}$ | $10^{-3}$ | 1 |
| SGDm | Lion, Scalar | 0.9 | - | 0.1 | 0.9 | - | - | - | $10^{-5}$ | $10^{-3}$ | 1 |

Table 3: The values of meta-parameters used in ImageNet dataset.

| Base Update | Meta Update (if any) | $\rho$ | $\lambda$ | $\kappa$ | $c$ | $\bar{\rho}$ | $\bar{\lambda}$ | $\bar{c}$ | $\alpha_0$ | $\eta$ | $\gamma$ |
|---|---|---|---|---|---|---|---|---|---|---|---|
| AdamW | Fixed stepsize | 0.9 | 0.999 | 0.1 | - | - | - | - | $10^{-5}$ | - | 1 |
| | Adam, Scalar | 0.9 | 0.999 | 0.1 | - | 0.9 | 0.999 | - | $10^{-6}$ | $10^{-3}$ | 1 |
| Lion | Fixed stepsize | 0.99 | - | 0.1 | 0.9 | - | - | - | $10^{-4}$ | - | 1 |
| | Lion, Scalar | 0.99 | - | 0.1 | 0.9 | 0.99 | - | 0.9 | $10^{-6}$ | $10^{-3}$ | 1 |

Table 4: The values of meta-parameters used in TinyStories dataset.

1), MetaOptimize (SGDm, Lion) and MetaOptimize (AdamW, Lion) have the best performance. Moreover, in the test accuracy (top1), these two combinations of MetaOptimze outperform other hyperparameter optimization methods and only AdamW with a handcrafted learning rate scheduler has a slightly better performance at the end of the training process.

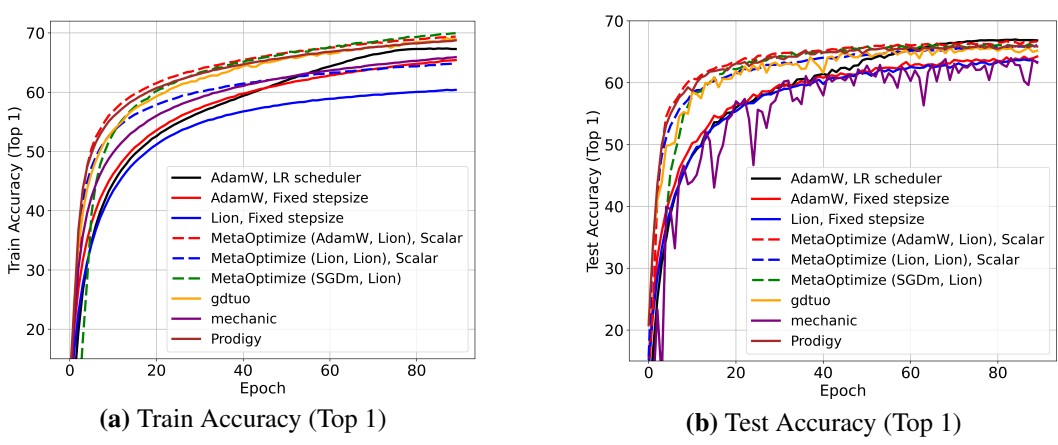

**(a)** Train Accuracy (Top 1)        **(b)** Test Accuracy (Top 1)

Figure 6: ImageNet learning curves.

In Figure 7, we provide the test loss of considered algorithms for the TinyStories datasets. As can be seen, the learning curves have the same trends as the training loss in Figure 5.

Figure 8 shows the results for the blockwise version of MetaOptimize for two combinations of (AdamW, Lion) and (Lion, Lion). As can be seen, they showed no improvement over the scalar version.

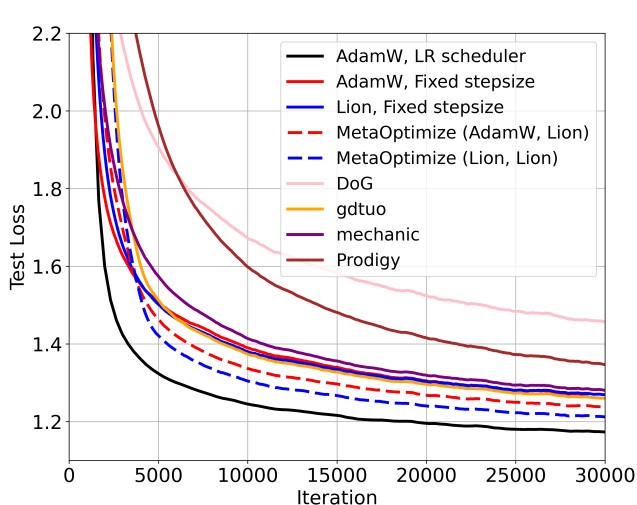

Figure 7: TinyStories learning curves.

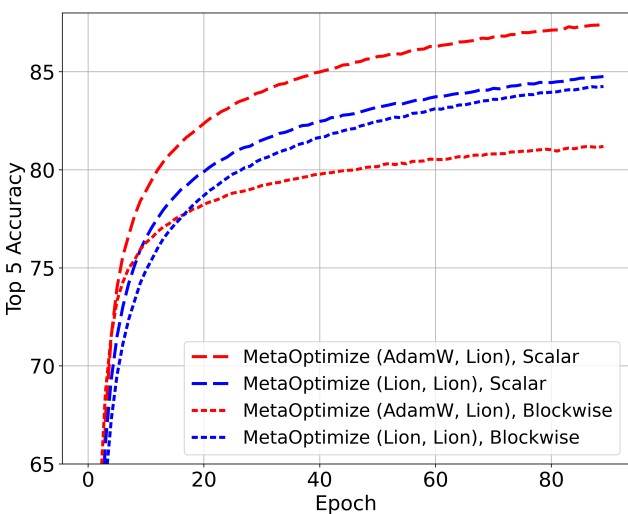

Figure 8: Comparison of blockwise version of MetaOptimize with the scalar version in ImageNet dataset.

