# OpenReview forum: "MetaOptimize: A Framework for Optimizing Step Sizes and Other Meta-parameters"
_ICLR.cc/2025/Conference — Submitted to ICLR 2025_

### Official Review · Reviewer_idAw · 2024-10-25

**Soundness:** 3
**Presentation:** 2
**Contribution:** 2
**Rating:** 5
**Confidence:** 2

**Summary:**

This paper proposes a MetaOptimize framework that dynamically adjusts hyperparameters (like step size) of base algorithms (like SGD and Adam), to minimize a discounted sum of future losses. Specifically, this work adopts an idea similar to eligibility traces in reinforcement learning to construct the future loss, uses base algorithm to minimize this future loss to output the weight vectors, leverages the weight vectors to compute the Jacobian matrix, and finally takes the Jacobian matrix as input of a meta algorithm to update the hyperparameters across different tasks/times. Complexity-reduced variants of MetaOptimize (include approximation version and Hessian-free version) are also proposed. Experiments on classification and language tasks are further conducted to validate the effectiveness of the proposed MetaOptimize framework. I did not check the proof line by line, because all the proofs in the appendix are about derivative computation, and such computation is basic.

**Strengths:**

(1)	This paper provides a novel MetaOptimize framework to meta-learn hyperparameters of base machine learning algorithms. In particular, the introduction of eligibility traces (from reinforcement learning) into the field of hyperparameter optimization is new.

(2)	The further proposed approximation version and Hessian-free version make great efforts to reduce the complexity of the MetaOptimize framework.

(3)	The proposed MetaOptimize framework in Algorithm 1 is general and really wraps around any first-order optimization algorithm.

(4)	Experiments on image classification and language modeling benchmarks are extensive, demonstrating the effectiveness of the proposed algorithm when compared with hand-crafted learning rate schedules.

**Weaknesses:**

**Major**

(1)	I admire that the introduction of eligibility traces into hyperparameter optimization is novel. However, when introducing an existing technique (from reinforcement learning) into your research problem (hyperparameter optimization), some theoretical or insightful analysis should be provided. For example, what is the insightful relationship between the objective in Eq.(5) and the future loss in Eq.(1)? Minimizing the objective in Eq.(5) can provably minimize the future loss in Eq.(1)? Such theoretical analysis is necessary, otherwise the current version in my opinion looks more like a direct application of eligibility traces (i.e. Eq.(5)) into hyperparameter optimization, hence resulting in limited insights.

(2)	The introduction of the Jacobian matrix $\mathcal{H} _{t}$ in line 222 is novel, but the bilevel-optimization (i.e. base-meta algorithm optimization) framework in Algorithm 1 is not new for me. Besides, it will be better to utilize existing theoretical analysis for bilevel programming (e.g. [1] and references therein) to derive specific theoretical results (e.g. the stability or generalization guarantees) for the proposed MetaOptimize framework in Algorithm 1.


**Minor**

(3)	There exists too many blank spaces in the paper. For example, between Eq.(10) and Eq.(11), I would suggest use $(\frac{\mathrm{d}y _{t+1}}{\mathrm{d} \beta _{\tau}}, \frac{\mathrm{d}x _{t+1}}{\mathrm{d} \beta _{\tau }}, \frac{\mathrm{d}h _{t+1}}{\mathrm{d} \beta _{\tau}})^{T} = G _{t}(\frac{\mathrm{d}y _{t}}{\mathrm{d} \beta _{\tau}}, \frac{\mathrm{d}x _{t}}{\mathrm{d} \beta _{\tau }}, \frac{\mathrm{d}h _{t}}{\mathrm{d} \beta _{\tau}})^{T}$ to save space, and give more theoretical explanations. Similar problem also exists in Eqs.(12)-(16).

(4)	Line 075, sum of future loss -> sum of future losses

(5)	Eq.(2) has a misleading typo: $x _{t} = Alg _{base} (x _{t}, \nabla f _{t}(w _{t}), \beta _{t})$?

(6)	Throughout the whole paper, the differential operator is $\mathrm{d}$ (mathrm{d}), not $d$.

(7)	Line 138, times $\tau > t$.) ->  times $\tau > t$).

(8)	Line 206, two case

(9)	Line 208, what does $|H _{t}|$ mean?

(10)	Line 234, $d,x _{t+1}/d, h _{t}$.

(11)	Line 318, zeroed out; -> zeroed out,


Reference

[1] Stability and Generalization of Bilevel Programming in Hyperparameter Optimization. NeurIPS2021.

**Questions:**

(1)	Can you give more theoretical explanations for why gradient descent in Eq.(5) can minimize the future loss defined in Eq.(1) ? Or can you give more theoretical analysis for the benefits of introduction of eligibility traces into hyperparameter optimization?

---

> ### Author Response · Authors · 2024-11-21
> **Author Response (part 1)**
>
> > (1) Can you give more theoretical explanations for why gradient descent in Eq.(5) can minimize the future loss defined in Eq.(1) ? Or can you give more theoretical analysis for the benefits of introduction of eligibility traces into hyperparameter optimization?
>
> We acknowledge that Equation (5) would benefit from additional intuition and theoretical support. Below, we provide further explanations and discuss potential alternatives.
>
> **Intuition Behind the Backward View Update (Equation 5):**
> The backward view update in Equation (5) serves as an approximation of the forward view update in Equation (4). Due to causality constraints, a backward view update is necessary because it allows us to reorder terms in time and apply them as soon as they become available. Although the terms in the two sums are the same, they differ in timing, enabling real-time updates.
>
> **Theoretical Backing in a Simple Regime:**
> The approximation becomes exact for sufficiently small meta-step sizes (i.e., when $\eta \to 0$). Specifically, consider some $T \geq 1$ and suppose that $f_t(\cdot) = 0$ for all $t < 0$ and all $t > T$. Then, as $\eta \to 0$, it can be shown that:
> $\frac{\beta^{(5)}_T-\beta_0}{\eta}\to \frac{\beta^{(4)}_T-\beta_0}{\eta}$,
> where $\beta^{(4)}_T$ and $\beta^{(5)}_T$ are the values of $\beta$ at time $T$ obtained from updates (4) and (5), respectively, starting from the same initial value $\beta_0$ at time $0$. Intuitively, this is because as $\eta\to0$, $\beta$ remains alomst constant over the intercal $[0,T]$, and the right hand side of (4) would be equal to the right hand side of (5) when summed over $[0,T]$, with accuracy $O(\eta^2)$. For larger values of $\eta$, the approximation may not be as accurate. This is on par with a similar result for eligibility traces in RL.
>
> **Alternative, More Accurate Solutions:**
> We recognize that developing a more accurate trace, similar to the Dutch traces in RL (defined in Chapter 11 of Sutton and Barto 2018), could improve performance. Dutch traces however work only for the linear case and  creating a more accurate trace for our context is a mathematically challenging problem that we have been working on since the submission. On the other hand, the simple eligibility trace is effective and widely used in RL algorithms like TD($\lambda$). On this grounds, we believe that the current simple trace serves as a good baseline, and we plan to explore more complex solutions in future work.
>
> **Modifications Applied to the Paper:**
> To address your comment, we have made the following revisions to the manuscript (updated pdf):
> 1. Indcluded the discussion of the approximation quality in the paragraph following Equation (5).
> 2. Added a discussion of possible alternative, more accurate solutions in the future works, in Section 9.
> 3. Enhanced the intuition behind the backward view update in the paragraph following Equation (5).
>
>
> > (2) The introduction of the Jacobian matrix $\mathcal{H}_t$ in line 222 is novel, but the bilevel-optimization (i.e. base-meta algorithm optimization) framework in Algorithm 1 is not new for me.
>
> While the base-meta optimization framework has roots in earlier works like IDBD (Sutton, 1982) and several base-meta optimization algorithms exist, we believe there is substantial room for improvement, especially when using blockwise step sizes. Algorithm 1 introduces significant differences from previous methods in the way we update the Jacobian matrix $\mathcal{H}_t$ based on Equations (10) and (15). It accounts for the dynamics of the meta-parameters $\beta$—specifically, how changes in the
> current $\beta$ affect future values of $\beta$.  Please refer to Remark 4.1 of the paper for detailed discussion.

---

> ### Author Response · Authors · 2024-11-21
> **Author Response (part 2)**
>
> > Besides, it will be better to utilize existing theoretical analysis for bilevel programming (e.g. [1] and references therein) to derive specific theoretical results (e.g. the stability or generalization guarantees) for the proposed MetaOptimize framework in Algorithm 1.
>
>
> Regarding the referenced paper [1], their analysis relies on unrolled differentiation (that unrolls the algorithm for several steps into future and approximates the derivative of future loss with respect to the current meta-paramaters), whereas MetaOptimize employs a causal algorithm where meta-updates depend only on past information. This fundamental difference makes it challenging to directly leverage their theoretical results for our framework.
>
> We acknowledge that theoretical guarantees are important and provide valuable insights. However, incorporating a comprehensive theoretical analysis is beyond the scope of our current work for two reasons: 1- The field of meta-parameter optimization is still evolving, and both the community and we, as authors, are exploring its full potential. It may be premature to seek comprehensive theoretical performance guarantees at this stage. 2-  Such analysis typically relies on strong assumptions, like convexity or smoothness, which do not align with our focus on developing scalable algorithm effective for large-scale networks.
>
> We plan to explore analytical tools from bilevel optimization literature, including those in [1] and its references, in future work to develop theoretical results specific to MetaOptimize, such as stability and regret bounds.
>
> > Minor: (3) There exists too many blank spaces in the paper. For example, between Eq.(10) and Eq.(11), I would suggest use $\left( \frac{d y_{t+1}}{d \beta_{\tau}}, \frac{d x_{t+1}}{d \beta_{\tau}}, \frac{d h_{t+1}}{d \beta_{\tau}} \right)^T = G_t \left( \frac{d y_t}{d \beta_{\tau}}, \frac{d x_t}{d \beta_{\tau}}, \frac{d h_t}{d \beta_{\tau}} \right)^T$ to save space, and give more theoretical explanations. Similar problem also exists in Eqs.(12)-(16).
>
> We have applied your suggested change in the revised version. This adjustment has resolved the blank spaces and improved the overall appearance of the paper.
>
>
> > (4-11) Typos:
>
> We thank the reviewer for their meticulous reading. We corrected all the typos.

---

> > ### Comment · Reviewer_idAw · 2024-11-22
> > **Response to the Rebuttal**
> >
> > Thanks authors for their detailed response. Many of my concerns (e.g. the insightful connection between Eq.(5) and Eq. (1)) have been addressed. However, given that the theoretical contribution (or say insightful analysis of the properties of proposed algorithm) is limited, I will maintain my initial score.
> >
> > Besides, I admire that authors can incorporate our reviews into the revision of this paper, but I would suggest authors to render the revised part (e.g. the correction of some typoes) in other color (like purple) to help us reviewers find the changes of the revised paper.
> >
> > Overall, I will keep my score.

---

> > > ### Author Response · Authors · 2024-12-02
> > >
> > > We thank the reviewer for their feedback and are glad that our proposed lemma on the equivalence of forward and backward views was found insightful. We conducted experiments that support some of the insights provided below, which also contributed to the delay in our response.
> > >
> > > >  Theoretical contribution and insightful analysis of the properties of proposed algorithm is limited
> > >
> > > We appreciate this valid concern, and we acknowledge that the initial version of our paper may have overlooked a thorough discussion of insights/analysis. Below, we provide additional insights and discussion on analysis of MetaOptimize properties, which we hope will address this concern.
> > >
> > > **Interpretation of MetaOptimize update rule:**
> > >
> > > The MetaOptimize update rules have an intuitive interpretation. Consider the Hessian-free version of MetaOptimize, presented in Algorithm 3 of the paper, for simplicity of the discussion. In this algorithm, after computing weight updates $\Delta w$ using the base-algorithm $w_{t+1}  \leftarrow w_t + \Delta w$, the meta-gradient is obtained as follows:
> > >
> > > $h_{t+1}= \gamma(1 -\kappa\alpha_t) h_t  + \Delta w$,
> > >
> > > meta-gradient $\simeq h_{t+1}^T \nabla f_t(w_t)$.
> > >
> > > Here, $h_t$ represents a trace of past $w$-updates,  and meta-gradient measures how well the recent evolution of $w$ is aligned with the gradient of $f_t$. If the correlation is negative, it indicates that the recent $w$-updates have resulted in a decrease of $f_t$, and the decrease would have been larger if stepsize was larger; therefore the meta-update increases the stepsize.  On the other hand a positive correlation indicates unreliable past $w$ updates (e.g., due to zigzagging), in which case the meta-update reduces the stepsize.
> > >
> > > The horizon (or decay rate) of the trace $h$ is controlled by $\gamma$. In Hessian-based MetaOptimize (without Hessian-free approximations), the decay rate, $\gamma(I-\alpha\nabla^2 f)$, is influenced also by the Hessian and step size.  In particular, as the gradients change more rapidly in a curved landscape, a large hessian helps to diminish the effect of older $\Delta w$ on $h$ more rapidly.
> > >
> > >
> > > **Insightful properties of MetaOptimize in our new continual-learning experiments:**
> > >
> > > In addition to our original contributions, we conducted experiments on continual-CIFAR100 benchmark, which involves training on sequential tasks, each consisting of unique CIFAR100 classes.  MetaOptimize consistently outperformed fixed step-size baselines, with blockwise variants achieving the best results. The details have been reported in Our response to Reviewer GWeq.
> > >
> > > Related to properties of MetaOptimize, some interesting stepsize-patterns emerged in these experiments, that highlight the strengths of meta-gradient updates in continual learning:
> > >
> > > - MetaOptimize revealed a sawtooth pattern, with stepsizes increasing at the start of each task to help adaptation to the new task, and decreasing toward the end of each task for fine-tuning.  Such behavior is challenging to replicate with hand-crafted schedules.
> > > - Blockwise stepsizes revealed distinct adaptation behaviors. Stepsizes for initial layers decreased across tasks (indicating generalization of the learned features), while final-layer stepsizes remained consistent across tasks (which is necessary for adapting to new tasks).
> > >
> > > **Core idea -- gradient decent on meta-paramters:**
> > >
> > > MetaOptimize provides a framework to estimate the meta-gradient, i.e., the gradient of the future discounted loss with respect to meta-parameters, and update the meta-parameters using optimizers like SGD or Adam. To maintain causalty, we incorporate backward-view approximation (i.e., eligibility trace) as discussed in the previous response. The distinction of MetaOptimize from its successor methods (such as IDBD and Hypergradient descent) is that MetaOptimize formulates an exact computation of the eligibility trace, taking intoaccount the impact of step-size dynamics, unlike prior approximations.
> > >
> > > **Theoretical performance guarantee:**
> > >
> > > If the backward-view approximation were exact, MetaOptimize would provably outperform fixed meta-parameters by moving downhill on the average future loss. However, this equivalence holds only in the asymptotically small $\eta$ regime (discussed in our previouse response). For practical meta-stepsizes $\eta>0$, the lack of equivalence makes deriving theoretical guarantees significantly more challenging.
> > >
> > > Developing more accurate backward-view traces, as discussed in the Limitations section, is key to achieving this equivalence. Once realized, theoretical guarantees would become significantly simpler to derive.
> > >
> > > The current paper sets the stage for this effort by introducing a systematic framework that decomposes the problem into manageable components. This foundation aims to guide future research toward a theoretically robust meta-optimization algorithm. In this spirit the paper has a quite comprehensive Limitations section.

---

> > > ### Author Response · Authors · 2024-12-02
> > >
> > > **Revisions for Final Paper:**
> > > We will incorporate the insights provided here—especially around the interpretation of MetaOptimize update rules, insights obtained from the continual-learning experiments, and the theoretical discussions—into the final paper revision.
> > >
> > > We appreciate your feedback and hope the insights shared here address your concerns. If there are additional points or suggestions, we would be glad to incorporate them into our future revisions. Thank you for your time and effort in reviewing our work.

---

### Official Review · Reviewer_GWeq · 2024-10-29

**Soundness:** 2
**Presentation:** 1
**Contribution:** 2
**Rating:** 5
**Confidence:** 3

**Summary:**

The submission aims to adapt the learning rate while model training. By minimising the regret of the historical model training, the learning rate is updated by the meta gradient computed by a proposed MetaOptimise framework wrapped with the proposed approximation method to reduce the computational cost, especially that caused by the Hessian matrix. The method is empirically texted on both image and language datasets with learning curve and computational cost shown as demonstration for efficiency.

**Strengths:**

1. The motivation that learning learning rate is clear and practical.
2. The operation details are discussed and given in the appendix for readers to have a broad insight.

**Weaknesses:**

1. The paper is not presented clearly.
2. The experiment results do not cover the generalisation performance of the trained models by different optimisations. For example, there is no direct comparison of the accuracy of the test set. The top 5 accuracy applied in the paper can reflect the performance of the optimisers but usually top 1 accuracy is more important.
3. As learning rate learning is usually a special case of optimiser learning, some previous works are missing for comparison [1,2,3].
4. The proposed backward-view update is similar to the forward mode differentiation (FMD) in [1] and the follow-up study on learning rate learning paper [2]. In addition, Equation 1 only contains the training information, the motivation for optimising the learning rate against this objective function is not clear and a follow-up question can be what benefits are achieved by optimising this.
5. I believe approximate Hessian with either diagonal matrix or block diagonal matrix is common in the precondition optimisation field, L-approximation is another special ly format which is parallel with the others. Stating it as containing existing algorithms as special cases may be overclaiming. Also, it could be interesting to know that with diagonal matrix approximation, the Meta-optimize works and it will definitely reduce computational cost.
6. Suggestions: The reference format is a little mass, which does not affect my score for this paper.  For example, DoG is published in ICML 2023 while in the reference it is the arXiv version.

[1] Khodak M, Balcan MF, Talwalkar AS. Adaptive gradient-based meta-learning methods. Advances in Neural Information Processing Systems. 2019;32.
[2] Micaelli P, Storkey AJ. Gradient-based hyperparameter optimization over long horizons. Advances in Neural Information Processing Systems. 2021 Dec 6;34:10798-809.
[3] Gao B, Gouk H, Lee HB, Hospedales TM. Meta mirror descent: Optimiser learning for fast convergence. arXiv preprint arXiv:2203.02711. 2022 Mar 5.

**Questions:**

1. In terms of time-space complexity comparison, MetaOptimize does not show a significant advantage on ImageNet but on TinyStoires. Can the authorise explain why?

Also, see the weaknesses section.

---

> ### Author Response · Authors · 2024-11-21
> **Author Response**
>
> We thank the reviewer for their feedback. We believe our revision has improved the paper and clarity of its presentation, as detailed below.
>
> > 2. The experiment results do not cover the generalization performance of the trained models by different optimizations. For example, there is no direct comparison of the accuracy of the test set. The top 5 accuracy applied in the paper can reflect the performance of the optimizers but usually top 1 accuracy is more important.
>
> **Modifications Applied to the Paper:**
> In the revised manuscript (updated pdf), we  report generalization performance, including test accuracy (for ImageNet) and test loss (for Language Modeling), in Appendix D, specifically focusing on top-1 accuracy. The test accuracies exhibit similar trends to the training accuracy curves and maintain the same relative performance among different algorithms. In all experiments, MetaOptimize outperforms all baselines in both training and test accuracies. In addition to test accuracies, we continue to report training accuracies in the revised paper.
>
> **Relevance of Train vs. Test Accuracies for Continual Learning:**
> Although our experiments involve stationary supervised learning benchmarks, we develop MetaOptimize within the increasingly popular continual learning framework, which is more challenging and more general compared to the classical supervised learning framework. In this continual setting, we handle a stream of potentially non-stationary loss functions rather than fixed train-test splits, and as a result the appropriate performance measure is the online loss (or regret) rather than test set performance. Furthermore, techniques for improving generalization under meta-parameter optimization—such as specific regularizations—may differ from conventional methods known to work well with fixed step sizes. This area warrants further research.
>
>
> > 3 & 4. As learning rate learning is usually a special case of optimiser learning, some previous works are missing for comparison [1,2,3].  The proposed backward-view update is similar to the forward mode differentiation (FMD) in [1] and the follow-up study on learning rate learning paper [2].
>
> Thank you for bringing these references to our attention. We have included them in the related works section of the revised manuscript and discussed their connections to MetaOptimize.
>
> **Regarding [1]:** The setting in Khodak et al. (2019) differs significantly from ours. They consider a sequence of tasks, each of length m, whereas in our continual learning framework, we assume m = 1 with no prior knowledge about task changes. Their analysis relies on the assumption that all loss functions are convex. Under this strong assumption, even vanilla SGD with fixed step sizes achieves average regret bounds of order $1/\sqrt{m}$. They proposed meta-algorithms that attain regret bounds of the same order but with improved constants. In our context, assuming convex loss functions with m = 1 would result in constant average regret, possibly with small constant-improvements.
>
> **Regarding [2]:** The algorithm in (Micaelli and Storkey 2021) is equivalent to the L-approximation, and is essentially similar to IDBD (Sutton 1992) and its extensions (Xu et al. 2018). Like IDBD, their approach ignores the impact of the dynamics of meta-parameters and considers only the dynamics of the base algorithms. We highlighted this difference in Remark 4.1 of our submitted manuscript. Additionally, we explicitly discuss the forward objective and its connection to the backward approximation, rather than treating them as two independent techniques.
>
> **Regarding [3]:** Gao et al. (2022) addressed a different setting involving multitask learning. Specifically, they update meta-parameters after a complete training on one of the tasks. This approach does not align with our experiments, which involve concurrent learning of step sizes and network weights. Moreover, as their approach converges to a fixed stepsize, the resulting baseline would perform no better than the best-fixed-stepsize baseline that is already outperformed by MetaOptimize in our experiments.
>
> **Our Choice of Baselines:** We have carefully selected state-of-the-art methods for comparison in our experiments. Including additional baselines would overcrowd the figures and make them difficult to interpret. However, we acknowledge the relevance of these works and have discussed their connections to MetaOptimize in the revised paper.

---

> ### Author Response · Authors · 2024-11-21
> **Author Response (part 2)**
>
> > In addition, Equation 1 only contains the training information, the motivation for optimising the learning rate against this objective function is not clear and a follow-up question can be what benefits are achieved by optimising this.
>
> Yes, optimizing the learning rate against this objective is beneficial, especially in the continual learning setting, which is more challenging and realistic than classical supervised learning, and which is emerging as a a standard paradigm for big-world learning. In continual learning, there are no fixed training or test sets; instead, data arrives as a potentially non-stationary stream. Minimizing the online loss (or regret) allows the model to adapt dynamically to new information.
>
> Interestingly, although MetaOptimize is designed for continual learning, we have demonstrated that it also outperforms existing baselines on classical supervised learning tasks. We agree that additional performance gains would be possible in supervised learning, by integrating validation data into the algorithm (e.g., into regularizers, or by using validaion-gradients in  $\beta$ updates, etc.). However, this is beyond the scope of current paper, and we do no want to limit the proposed algorithms to classical supervised learning.
>
>
> > 5. I believe approximate Hessian with either diagonal matrix or block diagonal matrix is common in the precondition optimisation field, L-approximation is another special ly format which is parallel with the others. Stating it as containing existing algorithms as special cases may be overclaiming.
>
> The L-approximation in our work is not merely a matrix approximation; it has a specific interpretation. By removing the upper-right block of the matrix G in the L-approximation, we effectively ignore the dynamics of the meta-parameter β. Therefore, algorithms that reduce to the L-approximation are indeed special cases of meta-parameter optimization methods that neglect the dynamics of β, as discussed in Section 5.
>
> Regarding Hessian approximation techniques like diagonal and block-diagonal approximations, while they are common in precondition optimisation, none of these methods provide the exact diagonal of the Hessian. Furthermore, incorporating Hessian would introduce several complexities beyond approximation quality and computational efficiency, even if we assume oracle access to the sample Hessian or its diagonal, as outlined in the Limitations section. More specifically, Hessian impacts the trace $H_{t+1}$ multiplicatively, making the updates sensitive to sample noise, which necessitated approproate clipping techniques. Additionally, the Hessian approximates the loss landscape's curvature but fails to account for non-differentiable curvatures, such as those from ReLU unit breakpoints, necessitating appropriate landscape-smoothing techiques. Exploring the impact of different Hessian approximations on MetaOptimize is beyond the scope of this work and warrants further exploration in future.
>
>
> > Also, it could be interesting to know that with diagonal matrix approximation, the Meta-optimize works and it will definitely reduce computational cost.
>
> While incorporating a diagonal matrix approximation could potentially reduce computational cost, our current version of MetaOptimize is Hessian-free and already exhibits very small computational overhead in both time and space, as reported in Table 1. We acknowledge that a low-complexity implementation of the Hessian might lead to performance improvements. However, implementing the Hessian—even in approximate form—introduces certain challenges beyond computational complexity, which we discussed in the Limitations section.
>
>
> >6. Suggestions: The reference format is a little mass, which does not affect my score for this paper. For example, DoG is published in ICML 2023 while in the reference it is the arXiv version.
>
>
> Thank you for bringing this to our attention. We have updated the reference formats in the revised manuscript, ensuring that all citations reflect their official conference or journal publications. For example, the DoG reference now cites its ICML 2023 version instead of the arXiv preprint. Please refer to the revised PDF for the updated references.
>
>
>
> > Questions: In terms of time-space complexity comparison, MetaOptimize does not show a significant advantage on ImageNet but on TinyStoires. Can the authorise explain why?
>
> The exact reason is not entirely clear, but it might be due to our sub-optimal implementation of MetaOptimize for the ImageNet experiments. We did not optimize our code for performance in this case, whereas for the baselines, we used the code provided by the original authors. As mentioned in the last paragraph of Appendix C, this could have affected the computational efficiency results.

---

> > ### Comment · Reviewer_GWeq · 2024-11-28
> >
> > I appreciate the authors’ feedback and detailed explanation regarding the benefits of the parameterization of L-approximation compared to Hessian-related methods, as well as the discussion about the suggested reference paper. This response has clarified the motivation and insights behind the authors’ approach. However, some questions remain, particularly regarding the performance of the proposed algorithm. For example, the reviewer notes that results on top-1 accuracy have been updated in Appendix D. Yet, due to the overlap of the learning curves across various optimizers, it is difficult to directly compare their generalization performance. Providing a table with accuracy metrics would help address this issue.
> > Additionally, while the authors argue that the primary focus of the optimizer is continual learning, the empirical justification provided is not comprehensive and does not adhere to standard benchmarks in the field. Based on the clarification provided, I have raised my score to 5 and am open to further discussion.

---

> > > ### Author Response · Authors · 2024-12-02
> > >
> > > Thank you for your thoughtful feedback and for raising your score. We appreciate your engagement and valuable suggestions, which have helped us refine our work further.
> > >
> > > >  due to the overlap of the learning curves across various optimizers, it is difficult to directly compare their generalization performance. Providing a table with accuracy metrics would help address this issue.
> > >
> > > We acknowledge this limitation. In response, we provide a table summarizing accuracy metrics from the ImageNet experiment, sorted by train accuracy for clarity:
> > >
> > >
> > > | Algorithm                    | Train Acc. (Top 1) | Test Acc. (Top 1) |
> > > |------------------------------|---------------|--------------|
> > > | MetaOptimize (SGDm, Lion)    | 69.96         | 66.01        |
> > > | MetaOptimize (AdamW, Lion)   | 69.38         | 66.66        |
> > > | gdtuo                        | 68.94         | 65.18        |
> > > | Prodigy                      | 68.75         | 65.85        |
> > > | AdamW, LR scheduler          | 67.29         | 66.84        |
> > > | MetaOptimize (Lion, Lion)    | 64.87         | 66.10        |
> > > | AdamW, Fixed stepsize        | 65.41         | 64.24        |
> > > | Mechanic                     | 65.92         | 63.22        |
> > > | Lion, Fixed stepsize         | 60.40         | 63.34        |
> > >
> > >
> > >
> > > We will report this table in the final version of the paper to facilitate comparisons.
> > >
> > >
> > > > Additionally, while the authors argue that the primary focus of the optimizer is continual learning, the empirical justification provided is not comprehensive and does not adhere to standard benchmarks in the field.
> > >
> > > Your point about the lack of comprehensive empirical justification for continual learning is entirely valid. To address this, we conducted experiments on continual-CIFAR100 benchmark, which involves training on sequential tasks, each consisting of unique CIFAR100 classes.
> > >
> > >
> > > **Experimental Setup:**
> > > - Tasks: Each task consists of 10 (or 20) randomly chosen classes from CIFAR100, trained for one epoch before moving to the next task. The total number of tasks is 100 divided by #classes-per-task.
> > > - Training Procedure: Each task was trained for one epoch, with batch size 1, to adhere to the continual learning paradigm.
> > > - Performance metric: Average online top-1 accuracy. This metric reflects generalization as each sample is seen only once.
> > > - Architecture: A small CNN with 2 convolutions (plus max-pooling) blocks followed by 2 fully connected layers (269K parameters).
> > >
> > >
> > > **Results:**
> > > MetaOptimize consistently outperformed fixed step-size baselines, with blockwise variants achieving the best results. The following tables summarize the average online top-1 accuracy averaged over 5 independent runs.
> > >
> > > Table 2: Average online top-1 accuracy (%) in continual CIFAR100 (10 classes per task):
> > > |   | AdamW  |  RMSProp | Lion  |
> > > |---|---|---|---|
> > > | Fixed step-size | 36.59 | 37.32 | 24.19 |
> > > | MetaOptimize (scalar) | 37.77 | 38.77 | 25.62 |
> > > | MetaOptimize (blockwise) | **39.05** | **39.79** | **26.4** |
> > >
> > > Table 3: Average online top-1 accuracy (%) in continual CIFAR100 (20 classes per task):
> > > |   | AdamW  |  RMSProp | Lion  |
> > > |---|---|---|---|
> > > | Fixed step-size | 24.84 | 21.81 | 14.45 |
> > > | MetaOptimize (scalar) | **26.24** | 24.9 | 15.19 |
> > > | MetaOptimize (blockwise) | 25.77 | **27.36** | **17.46** |
> > >
> > > In these tables, MetaOptimize employs the same algorithm for meta updates as the corresponding base algorithm; for instance, AdamW is used for both base updates and meta updates when the base algorithm is AdamW.
> > >
> > > **Other Findings:**
> > > - Stepsize Trends: MetaOptimize revealed a sawtooth pattern, with stepsizes increasing at the start of each task to help adaptation to the new task, and decreasing toward the end for fine-tuning
> > > - Stepsize across blocks: Blockwise stepsizes revealed distinct adaptation behaviors. Stepsizes for initial layers decreased across tasks (indicating generalization of the learned features), while final-layer stepsizes remained consistent across tasks (which is necessary for adapting to new tasks).
> > >
> > >
> > > We will add the results of this experiment to the final version of the paper. We appreciate your suggestion to strengthen this aspect of our study, and we welcome further discussion or feedback.

---

### Official Review · Reviewer_rEs8 · 2024-11-02

**Soundness:** 2
**Presentation:** 2
**Contribution:** 2
**Rating:** 5
**Confidence:** 3

**Summary:**

The paper provides a framework to dynamically update hyperparameters such as learning rate during training.

**Strengths:**

The proposed framework contains some existing algorithms as specials cases.

Their explanations are clear and facilitates easy understanding.

**Weaknesses:**

While the explanations provided are clear and easy to understand, some aspects are not thoroughly covered, e.g. Equation (5) would benefit from additional intuition or theoretical support.

There are established lines of work on gradient-based hyperparameter optimization using bilevel optimization. Discussing the connection between the proposed method and these existing approaches would strengthen the paper.

**Questions:**

Please discuss Equation (5) and the connection between the proposed method to bilevel optimization.

---

> ### Author Response · Authors · 2024-11-21
> **Author Response**
>
> >While the explanations provided are clear and easy to understand, some aspects are not thoroughly covered, e.g. Equation (5) would benefit from additional intuition or theoretical support.
>
> We acknowledge that Equation (5) would benefit from additional intuition and theoretical support. Below, we provide further explanations and discuss potential alternatives.
>
> **Intuition Behind the Backward View Update (Equation 5):**
> The backward view update in Equation (5) serves as an approximation of the forward view update in Equation (4). Due to causality constraints, a backward view update is necessary because it allows us to reorder terms in time and apply them as soon as they become available. Although the terms in the two sums are the same, they differ in timing, enabling real-time updates.
>
> **Approximation Quality:**
> The approximation becomes exact for sufficiently small meta-step sizes (i.e., when $\eta \to 0$). Specifically, consider some $T \geq 1$ and suppose that $f_t(\cdot) = 0$ for all $t < 0$ and all $t > T$. Then, as $\eta \to 0$, it can be shown that:
> $\frac{\beta^{(5)}_T-\beta_0}{\eta}\to \frac{\beta^{(4)}_T-\beta_0}{\eta}$,
> where $\beta^{(4)}_T$ and $\beta^{(5)}_T$ are the values of $\beta$ at time $T$ obtained from updates (4) and (5), respectively, starting from the same initial value $\beta_0$ at time $0$. Intuitively, this is because as $\eta\to0$, $\beta$ remains alomst constant over the intercal $[0,T]$, and the right hand side of (4) would be equal to the right hand side of (5) when summed over $[0,T]$, with accuracy $O(\eta^2)$. For larger values of $\eta$, the approximation may not be as accurate. This is on par with a similar result for eligibility traces in RL.
>
> **Alternative, More Accurate Solutions:**
> We recognize that developing a more accurate trace, similar to the Dutch traces in RL (defined in Chapter 11 of Sutton and Barto 2018), could improve performance. Dutch traces however work only for the linear case and  creating a more accurate trace for our context is a mathematically challenging problem that we have been working on since the submission. On the other hand, the simple eligibility trace is effective and widely used in RL algorithms like TD($\lambda$). On this grounds, we believe that the current simple trace serves as a good baseline, and we plan to explore more complex solutions in future work.
>
> **Modifications Applied to the Paper:**
> To address your comment, we have made the following revisions to the manuscript (updated pdf):
> 1. Indcluded the discussion of the approximation quality in the paragraph following Equation (5).
> 2. Added a discussion of possible alternative, more accurate solutions in the future works, in Section 9.
> 3. Enhanced the intuition behind the backward view update in the paragraph following Equation (5).
>
>
> >There are established lines of work on gradient-based hyperparameter optimization using bilevel optimization. Discussing the connection between the proposed method and these existing approaches would strengthen the paper.
>
> **Bilevel optimization and its distinctions from MetaOptimize:**
> Bilevel optimization in supervised learning aims to tune hyperparameters to minimize validation loss, typically involving an outer loop that updates hyperparameters after a complete training run of the base algorithm. In contrast, our proposed MetaOptimize framework is designed for continual learning scenarios where meta-parameters are optimized on-the-fly concurrently with training in a single streaming run. Unlike bilevel optimization methods that rely on a validation set—which is absent in continual learning—MetaOptimize focuses on minimizing the online loss (or regret) over a stream of data without separate validation or test sets. This makes MetaOptimize more appropriate for the increasingly popular setting of continual learning. We however left experiments on continual learning benchmarks for future works partly because benchmarking for continual learning is an evolving area, and there is still no standard large-scale benchmark available. Another advantage of online optimization of meta-parameters (as in MetaOptimize) compared to bilevel optimization, is the automatical learning of time-varying stepsizes and good stepsize schedules, which has a potential advantage over the fixed stepsizes learned through bilevel optimization, as demonstraited in the experiments (i.e., MetaOptimize outperforms best fixed stepsize).
>
> **Modifications Applied to the Paper:**
> In the revised manuscript, we have added a paragraph in the Related Works section to discuss bilevel optimization algorithms and clarify their connections and distinctions with MetaOptimize. This includes several works form the bilevel optimization literature, including the pioneering work of  (Bengio 2000), and its extensions in  (Maclaurin et al. 2015), (Pedregosa 2016),  (Franceschi et al. 2018), and  (Lorraine et al. 2020). Please refer to the updated PDF for details.

---

> ### Author Response · Authors · 2024-12-03
>
> We sincerely appreciate the time and effort you have devoted to reviewing our work and for sharing your valuable insights. We have carefully addressed the concerns and suggestions you raised and have provided detailed responses to each point.
>
> As the open discussion phase is nearing its conclusion, we would greatly appreciate any final feedback or thoughts you might have on our revisions and in light of our responses.
>
> Thank you again for your time and contributions to this process.

---

### Meta-Review · Area_Chair_ALCG · 2024-12-17

**Metareview:**

The paper introduces a method to dynamically optimize the learning rate of a machine learning algorithm. The method is based on eligibility trace and can wrap around any first-order optimization algorithm.

The reviewers in general agree that the paper is easy to follow, the introduction of eligibility trace to hyperparameter optimization is novel and interesting, and the method is general since it can wrap around any first-order optimization algorithm.

However, the reviewers also expressed concerns about some important weaknesses: the paper does not report generalization performance (which is important for machine learning algorithms), and the added generalization results are in general not satisfactory; the theoretical contribution of the paper may be limited, and hence the paper may be an application of eligibility trace to hyperparameter optimization which may limit its technical novelty.

Therefore, rejection is recommended.

**Additional Comments On Reviewer Discussion:**

During rebuttal and discussion phases, the reviewers expressed important concerns about the paper (as discussed above), which were not sufficiently addressed during rebuttal.

---

### Decision · Program_Chairs · 2025-01-22

Reject